# WHICH FORMAL LANGUAGES CAN LARGE LANGUAGE MODELS LEARN IN CONTEXT?

## ABSTRACT

In-context learning (ICL) drives much of the practical utility of large language models (LLMs), but its limitations—particularly on tasks requiring algorithmic reasoning—lack a precise characterization. Theoretically, transformer networks with unlimited chain-of-thought tokens in their output should be able to simulate any learning algorithm, but recent work has found that LLMs fall far short in practice. In this paper, we contribute to the growing body of work examining this discrepancy by evaluating the ICL capabilities of several LLMs (ChatGPT, DeepSeek, Qwen, and Llama) on a suite of formal language recognition tasks, which provide a controlled testbed for assessing reasoning ability grounded in the theory of computation. Our experiments span a range of language classes, namely regular, deterministic context-free, context-free, and context-sensitive languages. Bearing in mind recent work showing that a transformer network's expressive power increases with the number of padding tokens in its input, we test several ways of encoding exemplars that result in varying numbers of input tokens. To test the role of chain-of-thought, we also test prompts that require the model to produce an output immediately after reading the input and prompts that permit unrestricted reasoning before a label is produced. We find that pretrained LLMs perform very poorly on these reasoning tasks in all cases, only successfully learning the language of binary strings that begin with a 1. Also, contrary to expectation, adding padding and chain-of-thought tokens does not consistently improve accuracy. Still, ICL with pretrained LLMs is consistently more accurate than training a small transformer from scratch on the same data, suggesting that pretraining imbues transformers with a learning mechanism that is at least more sample efficient than training from scratch. These results reveal a disconnect between theoretical models of transformer capacity and the practical behavior of LLMs in ICL. Our code is publicly available.[1]

## 1 INTRODUCTION

Pretraining on massive amounts of data imbues transformer networks with a remarkable ability: without any parameter updates, the resulting large language model (LLM) can learn to solve novel tasks merely by observing examples of the task in its input. This ability, dubbed in-context learning (ICL, Brown et al., 2020), makes LLMs remarkably adaptable: with only a few demonstrations, they can perform tasks such as retrieval-augmented generation (Lewis et al., 2020), classification (Min et al., 2022), and code synthesis (Du et al., 2024). As a result, ICL has become a cornerstone of how LLMs are applied in practice, enabling a whole industry for prompt engineering.

Despite ICL's importance, we lack a clear understanding of conditions where it excels and when it fails. Successful ICL in LLMs hinges upon at least three factors: (1) the expressivity of the transformer architecture, on which LLMs are based; (2) the data and algorithm used to pretrain the LLM; and (3) the specific ICL algorithm the LLM has acquired from pretraining. The first point appears not to be at issue. (Merrill & Sabharwal, 2024) showed that, under certain simplifying assumptions, transformers with unlimited chain-of-thought steps are Turing-complete (Merrill & Sabharwal, 2024), so they should be able to execute any learning algorithm. Whether they learn these algorithms is another story; perhaps bigger and better pretraining results in better ICL in the

---

[1]See the anonymous supplementary material.

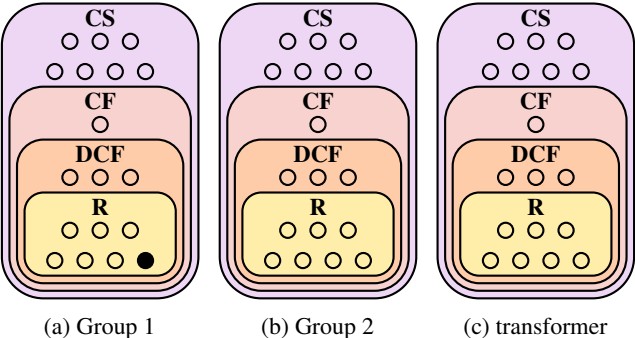

(a) Group 1      (b) Group 2      (c) transformer

Figure 1: Summary of our experimental results, shown in the same format as Butoi et al. (2025) for direct comparison. Dots denote individual languages; a filled dot indicates perfect (100%) held-out accuracy on that language. Languages are grouped by class—regular (**R**), deterministic context-free (**DCF**), context-free (**CF**), and context-sensitive (**CS**)—with panels (a)–(c) corresponding to the model groups labeled beneath each panel. The image shows (a) Group 1 indicates GPT-4o mini, DeepSeek-V3, Qwen2.5 32B, Llama-3.1 70B; (b) Group 2 indicates Qwen2.5 7B, Llama-3.1 8B and (c) baseline transformer trained on the same 100 examples.

limit. For now, however, LLMs falter on reasoning tasks that exceed a certain length or complexity threshold (Pfau et al., 2024; Bertsch et al., 2025; Li et al., 2024).

We investigate this discrepancy between the theoretical headroom and actual performance of ICL from a previously unstudied angle: in-context formal language recognition, a principled testbed grounded in the theory of computation. Formal language recognition tasks present a learner with a finite set of strings labeled as members or non-members of a mystery language. The model must learn to classify new strings accurately. We use an established language recognition benchmark, FLaRe (Butoi et al., 2025), which includes languages from varying levels of the Chomsky hierarchy, namely the classes of regular, deterministic context-free, context-free, and context-sensitive languages. This provides a benchmark ranked by computational difficulty.

Our study fixes the task while directly manipulating computation length. Each prompt contains 100 labeled string–label pairs from a language $L$ followed by one test string. We vary the input token budget by changing how many tokens encode each symbol (thus altering the number of input timesteps without changing the underlying string logic), and we vary the output token budget by comparing prompts that require immediate output to prompts that allow intermediate reasoning. This design isolates the effect of input/output token budgets from confounds such as task changes or data scale (contrast prior work that varies the number of examples or tasks; Li et al. (2024); Bertsch et al. (2025)). We evaluate on datasets subsampled from FLaRe (Butoi et al., 2025) and consider pretrained LLMs—ChatGPT, DeepSeek, Qwen, and Llama models of multiple sizes—alongside a transformer trained from scratch on the same examples.

Across languages, we measure classification accuracy and analyze how performance depends on input timesteps and prompting strategy. Three patterns emerge: (1) increasing input or output token budgets does not yield consistent gains in ICL accuracy, contrary to theoretical expectations; (2) pretrained LLMs are markedly more sample-efficient than transformers trained from scratch on the same data; and (3) consistent with recent theory (Li & Cotterell, 2025; Yang et al., 2024), strong performance is concentrated in languages lower in the hierarchy (see Figure 1). These observations elucidate where current ICL falls short and position formal languages as a precise lens for probing the algorithms LLMs execute at inference time.

## 2 RELATED WORK

A growing body of work investigates whether large language models can perform symbolic generalization in the in-context learning (ICL) setting. One line of research studies whether models can learn formal languages from positive examples alone. Akyürek et al. (2024), for example, train transformers from scratch on REGBENCH, a synthetic benchmark of probabilistic regular

languages. Their findings highlight the limitations of standard architectures in recovering the underlying automaton structure. In contrast, we evaluate pretrained LLMs across a wider range of formal languages—extending beyond regular—to ask whether symbolic reasoning abilities emerge from scale and pretraining alone.

Gupta et al. (2025) evaluate LLMs on novel regular languages defined by sampled DFAs, using sequence completion and transduction tasks. They find that models underperform simple n-gram baselines, suggesting weak generalization to unseen formal structures. In contrast, we study recognition: given examples from a fixed language, the model must judge whether a new string belongs to it, isolating its ability to detect and apply underlying structural rules.

Other work introduces datasets where the symbolic rules are provided directly. Petty et al. (2025) present RELIC, a benchmark of randomly generated context-free grammars in which models are given the grammar explicitly and must reason over its structure. This approach evaluates whether models can follow supplied symbolic rules. By contrast, our experiments omit such specifications, requiring models to infer structure from examples alone and making recognition performance a direct measure of inductive generalization.

Finally, related concerns about systematic generalization arise outside the domain of formal languages. Shojaee et al. (2025) study "Large Reasoning Models" (LRMs) on synthetic puzzles such as Tower of Hanoi and Blocks World, where problem complexity can be manipulated independently of logical structure. They find that LRMs sometimes overthink simple tasks, improve at moderate difficulty, but collapse under higher complexity, failing to follow algorithmic structure even with solution templates. Whereas their work probes reasoning traces in puzzles, our study situates evaluation in formal language recognition, where complexity is precisely characterized by the Chomsky hierarchy.

## 3 BACKGROUND

Let $\top$ and $\bot$ denote values of true and false, respectively, and let $\mathbb{B} \stackrel{\text{def}}{=} \{\top, \bot\}$. Let $\textvisiblespace$ be a visible representation of a space symbol. Throughout, we fix standard mathematical notation for strings and formal languages.

### 3.1 FORMAL LANGUAGES

A **symbol alphabet** $\Sigma$ is a non-empty finite set of elements called **symbols**. A **string** is a finite sequence of symbols. We denote the empty string as $\varepsilon$. For any two strings $\boldsymbol{u}, \boldsymbol{v}$, we denote their concatenation as $\boldsymbol{uv}$. For any string $\boldsymbol{w}$, we define $\boldsymbol{w}^0 \stackrel{\text{def}}{=} \varepsilon$ and $\boldsymbol{w}^k \stackrel{\text{def}}{=} \boldsymbol{w}^{k-1}\boldsymbol{w}$ for $k \geq 1$. A **language** (or **formal language**) is a (possibly infinite) set of strings. A **language class** is a set of languages. An alphabet may be treated as a language of strings of length one. For any two languages $L_1, L_2$, we define their concatenation as $L_1 L_2 \stackrel{\text{def}}{=} \{\boldsymbol{uv} \mid \boldsymbol{u} \in L_1, \boldsymbol{v} \in L_2\}$. For any language $L$, we define $L^0 \stackrel{\text{def}}{=} \{\varepsilon\}$ and $L^k \stackrel{\text{def}}{=} L^{k-1}L$ for $k \geq 1$, and we denote $L^* \stackrel{\text{def}}{=} \bigcup_{k=0}^{\infty} L^k$ and $L^+ \stackrel{\text{def}}{=} LL^*$. We denote the length of string $\boldsymbol{w}$ as $|\boldsymbol{w}|$, the $i^{\text{th}}$ symbol of $\boldsymbol{w}$ as $w_i$, and the substring of $\boldsymbol{w}$ starting at position $i$ as $\boldsymbol{w}_{\geq i} \stackrel{\text{def}}{=} w_i \cdots w_n$.

Every decision problem can be cast as a formal language. Given an input alphabet $\Sigma$, the problem corresponds to the set of strings in $\Sigma^*$ for which the answer is "yes." Solving the problem then amounts to recognizing the corresponding language. A broader discussion appears in App. A.

### 3.2 LARGE LANGUAGE MODELS

LLMs generally consist of a causally masked transformer decoder[2] (Vaswani et al., 2017). In practice, they are most often used as black boxes: the user provides a prompt, and the model produces an output string sampled from a distribution conditioned on that prompt. This usage applies whether the model is accessed through a hosted API or deployed locally as an open-weight model—the underlying parameters remain fixed in either case. What differs is the level of visibility: some interfaces hide details such as tokenization, detokenization, or system prompts, while local deployments may expose

---

[2]Also commonly referred to as a "decoder-only" architecture.

these internals. In all cases, however, prompting is the mechanism of interaction, and weights are not updated during use.

Let $\Gamma$ denote the **LLM character alphabet**, the set of individual characters (e.g., Unicode characters or bytes) that the tokenizer operates on.

**Definition 1.** *A **large language model (LLM)** $M$ over $\Gamma$ is a **stochastic map** $M \colon \Gamma^* \rightsquigarrow D(\Gamma^*)$, i.e., a function from strings in $\Gamma^*$ to probability distributions over $\Gamma^*$. The input to an LLM is called a **prompt**.*

Suppose we have a number of labeled examples from a language over alphabet $\Sigma$ and a test string we want an LLM to classify. In order to feed them as a prompt to the LLM, we need a way of encoding them as a string over $\Gamma$.

**Definition 2.** *A **prompt template** $\tau$ from alphabet $\Sigma$ to alphabet $\Gamma$ is an invertible[3] function $\tau \colon (\Sigma^* \times \mathbb{B})^+ \times \Sigma^* \to \Gamma^*$. If $\tau(((\boldsymbol{x}^{(1)}, y^{(1)}), \ldots, (\boldsymbol{x}^{(n)}, y^{(n)})), \boldsymbol{x}) = \boldsymbol{p}$, we call $((\boldsymbol{x}^{(1)}, y^{(1)}), \ldots, (\boldsymbol{x}^{(n)}, y^{(n)}))$ the **examples**, $\boldsymbol{x}$ the **test string**, and $\boldsymbol{p}$ the **prompt**.*

When defining a particular prompt template, we will need a way of encoding a string of symbols in $\Sigma$ into a string of characters in $\Gamma$.

**Definition 3.** *A **string encoder** $g$ from alphabet $\Sigma$ to alphabet $\Gamma$ is a function $g \colon \Sigma^* \to \Gamma^*$.*

Each prompting setup must also have a way of parsing an accept or reject result from the output of the LLM.

**Definition 4.** *An **output decoder** $h$ from alphabet $\Gamma$ is a function $h \colon \Gamma^* \to \mathbb{B} \cup \{\emptyset\}$, where $\emptyset$ indicates an unintelligible result.*

## 4 METHOD

Let $L$ be a formal language over the symbol alphabet $\Sigma$, and let $M$ be an LLM over character alphabet $\Gamma$. We assume a training set consisting of a sequence of $n$ string–label pairs $D_{\text{train}} = ((\boldsymbol{x}^{(1)}, y^{(1)}), \ldots, (\boldsymbol{x}^{(n)}, y^{(n)}))$ and a test set consisting of a multiset of $m$ string–label pairs $D_{\text{test}} = \{(\boldsymbol{x}^{(1)}, y^{(1)}), \ldots, (\boldsymbol{x}^{(m)}, y^{(m)})\}$. Let $\tau$ denote a prompt template and $h$ an output decoder. We evaluate $M$ on $D_{\text{test}}$ by prompting it with $D_{\text{train}}$ and comparing predicted labels with gold labels:

1. For each test pair $(\boldsymbol{x}, y) \in D_{\text{test}}$:
   (a) Construct a prompt $\boldsymbol{p} = \tau(D_{\text{train}}, \boldsymbol{x})$.
   (b) Sample $\boldsymbol{y_p} \sim M(\boldsymbol{p})$.
   (c) Derive a predicted label $h(\boldsymbol{y_p})$.
   (d) Compare $h(\boldsymbol{y_p})$ with $y$.
2. Compute the accuracy of the model as the proportion of test strings in $D_{\text{test}}$ that are classified correctly.

### 4.1 STRING ENCODING STRATEGIES

LLMs operate on sequences of tokens drawn from a finite vocabulary rather than directly on the symbols of a formal language. A **tokenizer** maps a string over the character alphabet $\Gamma$ to a sequence of tokens from the **token vocabulary** $\Delta$. The length of this token sequence determines the input budget, i.e., the number of timesteps available for processing the string. To manipulate the number of timesteps while holding the underlying symbol string fixed, we define three encoding strategies: (1) many symbols are mapped to one token (many $\to$ one); (2) one symbol is mapped to one token (one $\to$ one); (3) one symbol is mapped to many tokens (one $\to$ many).

Each **encoding strategy** $g_x \colon \Sigma^* \to \Gamma^*$ involves defining a **symbol encoding** function $\bar{g}_x \colon \Sigma \to \Gamma^*$, a bijective mapping from the symbol alphabet to strings over the LLM's character alphabet $\Gamma$. To reduce the risk of pretraining leakage—for instance, an LLM may have encountered the parity language with

---

[3]Invertibility ensures that the prompt encodes the exemplars losslessly and is not a trivial template like $\tau(((\boldsymbol{x}^{(1)}, y^{(1)}), \ldots, (\boldsymbol{x}^{(n)}, y^{(n)})), \boldsymbol{w}) = \varepsilon$.

Table 1: Examples of symbol-to-token encoding strategies used in our experiments. Each row shows how the input string $\boldsymbol{w} = \texttt{0100}$ is encoded using a different technique. In the examples, $\bar{g}_x(\texttt{0}) = \boxed{\texttt{␣a}}$, $\bar{g}_x(\texttt{1}) = \boxed{\texttt{␣b}}$, $t = 2$, and $\delta = \boxed{\texttt{␣p}}$ (in the many $\to$ one strategy the symbols are mapped directly to single characters without the leading space).

| Encoding Strategy | Encoded String | Tokenized String |
|---|---|---|
| **many $\to$ one** | `abaa` | `aba` `a` |
| **one $\to$ one** | `a b a a` | `a` `␣b` `␣a` `␣a` |
| **one $\to$ many** | `a p b p a p a p` | `a` `␣p` `␣b` `␣p` `␣a` `␣p` `␣a` `␣p` |

the alphabet $\{\texttt{0},\texttt{1}\}$ in computer science textbooks—and to avoid introducing semantic biases, we sample the mapping from symbols to token strings at random.

We assume the LLM uses byte-pair encoding (BPE) tokenization. We assume that whitespace appears only at the beginning of tokens, so that token boundaries always align with space characters.

Let $\Delta' = \Delta \cap \{\texttt{a}, \ldots, \texttt{z}\}$ denote the set of single-character tokens consisting of a lowercase ASCII letter. Similarly, let $\Delta'' = \Delta \cap \{\texttt{␣}\}\{\texttt{a}, \ldots, \texttt{z}\}^+$ denote the set of tokens in $\Delta$ that begin with an ASCII space followed by one or more lowercase ASCII letters.

Table 1 illustrates these three strategies with a toy example before we describe each in detail.

### 4.1.1 MANY SYMBOLS, ONE TOKEN

We define a bijective mapping $\bar{g}_x \colon \Sigma \to \Delta'$ that maps symbols to distinct single-character tokens, and concatenate the resulting characters:

$$g_x(\boldsymbol{w}) \stackrel{\text{def}}{=} \bar{g}_x(w_1)\bar{g}_x(w_2)\cdots\bar{g}_x(w_n) \tag{1}$$

No spaces are inserted between encoded symbols, potentially allowing multiple symbols to be tokenized as a single token, depending on the LLM's tokenizer and character patterns. Because the mapping is restricted to single characters, the strategy can represent at most $|\Delta'|$ distinct symbols (26 in our experiments).

### 4.1.2 ONE SYMBOL, ONE TOKEN

To avoid semantic interference between the tokens reserved for the prompt template and those used to encode the input strings, we exclude from $\Delta''$ any token that appears in the template. We then define a bijective mapping $\bar{g}_x \colon \Sigma \to \Delta''$ that assigns each symbol to a distinct token. The encoded string is constructed by detokenizing and concatenating the mapped tokens, omitting the initial leading space (already supplied by the prompt template):

$$g_x(\boldsymbol{w}) \stackrel{\text{def}}{=} \left(\bar{g}_x(w_1)\cdots\bar{g}_x(w_n)\right)_{\geq 2} \tag{2}$$

Since each symbol's encoding begins with a space and corresponds exactly to a full token, the tokenizer is guaranteed to tokenize each symbol $w_i$ as $\bar{g}_x(w_i)$.

### 4.1.3 ONE SYMBOL, MANY TOKENS

To increase the number of inference steps per input symbol, we encode each symbol as a sequence of $t$ tokens. We define a bijective mapping $\bar{g}_x \colon \Sigma \to \Delta''$, where $\Delta''$ is the set of tokens starting with a space and not used in the prompt template. Additionally, we sample a distinct filler token $\delta \in \Delta''$, disjoint from the range of $\bar{g}_x$, to pad each symbol's encoding.

Each symbol $w_i$ is encoded by concatenating its unique token $\bar{g}_x(w_i)$ with $t-1$ copies of the filler token $\delta$. After concatenation, we drop the initial leading space, since it is already provided by the

prompt template.

$$g_x(\boldsymbol{w}) \stackrel{\text{def}}{=} \left( \bar{g}_x(w_1) \cdot \delta^{t-1} \cdots \bar{g}_x(w_n) \cdot \delta^{t-1} \right)_{\geq 2} \tag{3}$$

This encoding guarantees that each input symbol corresponds to exactly $t$ tokens, ensuring uniform inference depth while preserving symbol-level information.

## 4.2 PROMPTING STRATEGIES

Large language models are sensitive to prompt design: the structure and format of demonstrations influence model behavior (Min et al., 2022). Prompting strategies can therefore be used not only to control formatting effects but also to vary the number of output timesteps, enabling tests of the theoretical claim that chain-of-thought prompting broadens the range of tasks models can solve (Pfau et al., 2024). We study two prompting templates: an **immediate-output** prompt, which requires the model to output only the final label, and a **zero-shot reasoning** prompt, which encourages the model to reason step by step before producing its final answer (Kojima et al., 2022). The corresponding templates are illustrated in Figure 2. For an example prompt and corresponding model output for both prompt templates, see App. E.

Let $g_x$ denote the string encoding function defined in §4.1. We also define a label encoding function $g_y : \mathbb{B} \to \Sigma$ that maps the truth value of a proposition to a symbol:

$$g_y(\phi) \stackrel{\text{def}}{=} \begin{cases} \texttt{1} & \text{if } \phi, \\ \texttt{0} & \text{otherwise.} \end{cases} \tag{4}$$

### 4.2.1 IMMEDIATE OUTPUT

This template instructs the LLM to output only `0` or `1`, indicating whether the test string belongs to the target language. To enforce the immediate-output constraint, we restrict valid outputs to either a bare digit (0 or 1) or the same digit followed by a trailing dot. The output decoder $h$ is therefore:

$$h(\boldsymbol{y}) \stackrel{\text{def}}{=} \begin{cases} \top & \text{if } \boldsymbol{y} = \texttt{1} \text{ or } \boldsymbol{y} = \texttt{1.} \\ \bot & \text{if } \boldsymbol{y} = \texttt{0} \text{ or } \boldsymbol{y} = \texttt{0.} \\ \emptyset & \text{otherwise.} \end{cases} \tag{5}$$

### 4.2.2 ZERO-SHOT REASONING

This template permits the model to generate intermediate reasoning steps before the final prediction. To account for these intermediate steps, we extract the model's prediction as the last binary digit (`0` or `1`) appearing in the output string $\boldsymbol{y}$, denoted by d. The decoder is defined as:

$$h(\boldsymbol{y}) \stackrel{\text{def}}{=} \begin{cases} \top & \text{if } \text{d} = \texttt{1}, \\ \bot & \text{if } \text{d} = \texttt{0}, \\ \emptyset & \text{otherwise.} \end{cases} \tag{6}$$

## 5 EXPERIMENTS

We probe the ICL capabilities of LLMs through the lens of formal language recognition, following the FLaRe framework of Butoi et al. (2025). The benchmark covers languages from across the Chomsky hierarchy—regular, deterministic context-free, context-free, and context-sensitive—summarized in App. B. For each language, we construct a prompt with 100 labeled examples and evaluate on 100 test strings. To keep costs and context length manageable while maintaining balance, the prompt's examples are the first 50 positive and first 50 negative instances from the corresponding FLaRe split, randomly shuffled, thereby matching the distribution of the original FLaRe datasets; additional sampling details are provided in App. D. In particular, for each experiment we use the same training sets $D_{\text{train}}$ and test sets $D_{\text{test}}$ and ensure that the examples in $D_{\text{train}}$ are presented in the same order. Each test string is appended to the fixed 100-example prompt according to the prompt template and evaluated independently. Outputs are parsed using the decoders defined in §4.2, and predictions are

```
Here are some positive and
negative examples of strings
in a formal language. Each
example is on a separate line
and follows the format "label:
string", where label is 0 or 1
(meaning negative or positive,
respectively), and string is a
sequence of symbols separated by
spaces.
A blank line indicates the end
of the examples.
```
$g_y(y^{(1)})$: $g_x(\boldsymbol{x}^{(1)})$

$\vdots$

$g_y(y^{(n)})$: $g_x(\boldsymbol{x}^{(n)})$

```
Given these examples, predict
the label (either 0 or 1) of the
string after the text "string: "
on the following line. If there
is no text after "string: ", it
represents the empty string. Do
not provide any explanation, but
respond only with the label 0 or
1. Do not use special markup for
the label; just use the text 0
or 1.
string:
```
$g_x(\boldsymbol{x})$

(a) immediate output

```
Here are some positive and
negative examples of strings
in a formal language. Each
example is on a separate line
and follows the format "label:
string", where label is 0 or 1
(meaning negative or positive,
respectively), and string is a
sequence of symbols separated by
spaces.
A blank line indicates the end
of the examples.
```
$g_y(y^{(1)})$: $g_x(\boldsymbol{x}^{(1)})$

$\vdots$

$g_y(y^{(n)})$: $g_x(\boldsymbol{x}^{(n)})$

```
Given these examples, predict
the label (either 0 or 1) of the
string after the text "string: "
on the following line. If there
is no text after "string: ",
it represents the empty string.
Feel free to think step-by-step
about the problem, but at the
end of your response, respond
only with the label 0 or 1. Do
not use special markup for the
label; just use the text 0 or 1.
string:
```
$g_x(\boldsymbol{x})$

(b) zero-shot reasoning

Figure 2: Prompt templates for the immediate output and the zero-shot reasoning strategies. We write literal text in `black typewriter font` and dynamic content that we insert in the prompt in orange. We insert dynamically the string encodings of the examples and their labels.

compared to ground-truth labels. Following Butoi et al. (2025), we consider a language recognized if the model achieves 100% accuracy on the membership task. In addition to the two main prompting strategies described in §4.2, we also experimented with minor instruction and formatting variants, deferring details and ablations to D and E.

Our evaluation covers a diverse set of state-of-the-art models: DeepSeek-V3, GPT-4o mini, Qwen2.5 32B, Qwen2.5 7B, Llama-3.1 8B, and Llama-3.1 70B. The first two are accessed via their vendor APIs, while the others are executed on a compute cluster. More implementation details, including decoding parameters and hardware, are provided in App. D. To contextualize model performance, we additionally compare against a small transformer trained from scratch on the same 100 labeled examples that appear in the prompts. This design holds the recognition task constant while varying input and output token budgets, enabling us to connect empirical ICL behavior to theoretical predictions about computation under limited budgets.

## 6 RESULTS

We organize our results around three questions: (i) can LLMs reliably recognize formal languages in the ICL setting, (ii) how does varying the input token budget affect performance, and (iii) how does allowing more output tokens through reasoning prompts influence recognition accuracy.

Table 2 compares three settings: LLMs evaluated with 100 in-context examples, a transformer trained from scratch on the same 100 examples, a comparable transformer trained on 10k examples as reported by Butoi et al. (2025). As expected, the supervised transformer benefits substantially

Table 2: Accuracy of LLMs and transformers trained from scratch ("Tf") on the FLaRe benchmark (Butoi et al., 2025). For each LLM, we report the best score across prompting and string encoding strategies. Columns are grouped by the number of training examples. For comparison, we include results from Butoi et al. (2025), where a transformer is trained on 10k examples and evaluated on the full test set. For each language, we bold the best 100-example model, and also bold the 10k-example transformer if it achieves the best score.

| Class | Language | 100 examples | | | | | | | 10k examples |
| | | DeepSeek-V3 | GPT-4o mini | Qwen2.5 32B | Qwen2.5 7B | Llama-3.1 8B | Llama-3.1-70B-Instruct | Tf | Tf |
|---|---|---|---|---|---|---|---|---|---|
| R | Even Pairs | 0.59 | 0.59 | 0.58 | 0.54 | 0.53 | **0.61** | 0.60 | **1.00** |
| | Repeat 01 | **0.97** | 0.79 | 0.74 | 0.62 | 0.54 | 0.77 | 0.64 | 0.86 |
| | Parity | 0.58 | **0.66** | 0.57 | 0.51 | 0.51 | 0.50 | 0.52 | 0.60 |
| | Cycle Navigation | 0.83 | **0.84** | 0.83 | 0.79 | 0.56 | 0.83 | 0.83 | **0.93** |
| | Modular Arithmetic | 0.75 | 0.68 | **0.79** | 0.63 | 0.52 | 0.73 | 0.66 | **0.88** |
| | Dyck-$(2, 3)$ | 0.68 | 0.72 | **0.76** | 0.72 | 0.53 | 0.73 | 0.70 | **0.82** |
| | First | **1.00** | **1.00** | **1.00** | 0.76 | 0.52 | **1.00** | 0.94 | **1.00** |
| DCF | Majority | 0.85 | 0.79 | 0.88 | 0.67 | 0.53 | 0.74 | **0.91** | **1.00** |
| | Stack Manipulation | **0.76** | 0.72 | **0.76** | 0.75 | 0.51 | 0.74 | 0.69 | **0.87** |
| | Marked Reversal | 0.69 | 0.69 | **0.70** | 0.58 | 0.51 | 0.67 | **0.70** | **0.87** |
| CF | Unmarked Reversal | 0.54 | 0.53 | 0.52 | 0.55 | 0.55 | 0.53 | **0.56** | **0.63** |
| CS | Marked Copy | **0.79** | 0.71 | 0.78 | 0.64 | 0.52 | 0.74 | 0.69 | **0.86** |
| | Missing Duplicate | **0.76** | 0.73 | 0.74 | **0.76** | 0.51 | 0.75 | 0.72 | **0.86** |
| | Odds First | **0.75** | 0.69 | 0.69 | 0.66 | 0.52 | 0.69 | 0.64 | **0.86** |
| | Binary Addition | **0.80** | **0.80** | 0.77 | 0.69 | 0.53 | 0.77 | 0.75 | **0.88** |
| | Binary Multiplication | 0.77 | 0.76 | 0.77 | 0.72 | 0.62 | 0.77 | **0.79** | **0.92** |
| | Compute Sqrt | **0.80** | 0.74 | 0.76 | 0.72 | 0.56 | 0.75 | 0.72 | **0.86** |
| | Bucket Sort | **0.70** | 0.67 | 0.67 | 0.67 | 0.52 | 0.67 | 0.69 | **0.88** |

from additional data, but LLMs with only 100 in-context examples consistently outperform the 100-example baseline and in several cases approach or surpass the 10k-example results. The strongest cases are Repeat 01 and Parity, where ICL with many-to-one encoding technique outperforms the transformer trained with 10k-example. A detailed table with all scores is provided in Appendix G.1.

From the model-level comparison, several patterns emerge. Larger models consistently outperform smaller ones, especially on context-free and context-sensitive languages. Proprietary models generally achieve stronger results than open-weight models, but within the open-weight families, Qwen2.5 32B often matches or exceeds the performance of Llama-3.1 70B despite its smaller scale. By contrast, smaller open-weight models such as Qwen2.5 7B and Llama-3.1 8B perform very poorly across the benchmark. The only perfect accuracy we observed is by DeepSeek-V3, GPT-4o mini, Qwen2.5 32B, and Llama-3.1 70B on First. These results align with theoretical findings that transformers with softmax attention can recognize at most a subset of regular languages, of which First is a member (Li & Cotterell, 2025).

We also ran a set of prompt tuning experiments to test whether lightweight adaptation could improve recognition; detailed results are provided in Appendix F.

Table 3 shows that input compression is consistently beneficial: many-to-one encodings yield the highest accuracies for most models, while one-to-many encodings degrade sharply as the expansion factor grows. Output reasoning provides only modest gains at best, and in several cases reduces performance compared to immediate-output prompting. Small models such as Llama-3.1 8B remain insensitive to these variations, with performance close to chance level; this arises both from a high rate of unparsable outputs and from consistently producing the same label, as discussed in Appendix D. We also conduct a failure analysis of accuracy as a function of string length and find no systematic, length-dependent pattern; see Appendix G.3.

# 7 CONCLUSION

Our study provides empirical support for recent theoretical results showing that transformers with softmax attention are limited in their expressivity to a strict subclass of the regular languages. While large pretrained LLMs achieve high accuracy on several tasks, including some beyond this class, these successes appear to be approximations rather than evidence of broader computational power.

Table 3: Average accuracy across languages in FLaRe achieved by LLMs for each configuration of encoding strategy and prompting strategy. For the one-to-many (**one → many**), we show the accuracy when we vary the number of timesteps $t$, for $t = 2, \ldots, 5$.

| Model | Prompting Strategy | Encoding Strategy | | | | | |
|---|---|---|---|---|---|---|---|
| | | many → one | one → one | one → many | | | |
| | | | | 2 | 3 | 4 | 5 |
| DeepSeek-V3 | immediate output | **0.71** | 0.70 | 0.69 | 0.66 | 0.67 | 0.66 |
| | zero-shot reasoning | **0.70** | 0.68 | 0.66 | 0.68 | 0.67 | 0.66 |
| GPT-4o mini | immediate output | 0.58 | **0.68** | 0.67 | 0.65 | 0.62 | 0.61 |
| | zero-shot reasoning | 0.58 | 0.66 | **0.67** | 0.66 | 0.58 | 0.56 |
| Llama-3.1 70B | immediate output | **0.68** | 0.65 | 0.65 | 0.66 | 0.63 | 0.64 |
| | zero-shot reasoning | 0.61 | 0.58 | **0.63** | 0.60 | 0.60 | 0.62 |
| Llama-3.1 8B | immediate output | **0.50** | **0.50** | **0.50** | **0.50** | 0.49 | 0.48 |
| | zero-shot reasoning | **0.51** | **0.51** | 0.50 | 0.50 | 0.50 | 0.50 |
| Qwen2.5 32B | immediate output | 0.71 | 0.68 | 0.68 | 0.43 | 0.46 | 0.33 |
| | zero-shot reasoning | **0.72** | 0.70 | 0.63 | 0.64 | 0.61 | 0.50 |
| Qwen2.5 7B | immediate output | **0.64** | 0.51 | 0.54 | 0.49 | 0.54 | 0.53 |
| | zero-shot reasoning | **0.64** | 0.54 | 0.55 | 0.54 | 0.54 | 0.55 |

At the same time, our experiments reveal a gap between theoretical predictions and practical outcomes regarding timesteps. In theory, increasing the number of input or output timesteps should expand computational power. In practice, however, longer input budgets do not consistently yield higher accuracy, and extending output length through reasoning prompts often brings little benefit or even degrades performance.

This calls for further theoretical work to better reconcile the expressivity of transformers in principle with the behaviors observed in large pretrained models.

## ETHICS STATEMENT

This paper investigates the behavior of large language models on synthetic formal-language tasks and introduces a small transformer trained from scratch. No human subjects, personally identifiable information, or proprietary datasets are used. Inputs and labels are programmatically generated from precisely specified grammars or automata; thus, privacy risks are negligible, and no demographic attributes are collected or inferred.

We run instruction-tuned open models (Qwen2.5 and Llama 3.1 variants) and, where noted, closed models via public APIs only to evaluate recognition of formal languages. While foundation models may encode societal biases, the tasks in this work are deliberately decontextualized (purely symbolic), minimizing the risk of harmful content generation. We followed model-license terms and standard responsible-use guidelines.

## REPRODUCIBILITY STATEMENT

All materials to reproduce our results are included: code to generate datasets, code to generate tables and code to subsample FLaRe dataset and evaluate instruction-tuned checkpoints via `vLLM` with transformers; fixed random seeds and decoding settings; prompt templates with the exact chat formatting; and experiment logs with per-sequence outputs and aggregate metrics. We also provide scripts/configs to query closed-model APIs (DeepSeek-V3, GPT-4o mini) with the corresponding dates/settings. Model identities (Hugging Face repositories) and implementation specifics are summarized in App. D.

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

## A    DECISION PROBLEMS AND FORMAL LANGUAGES

Any (yes/no) decision problem can be represented as a formal language. Fix a finite alphabet $\Sigma$ and a reasonable encoding enc of problem instances as strings in $\Sigma^*$. The problem then corresponds to

$$L = \left\{ x \in \Sigma^* : x = \text{enc}(I) \text{ for some instance } I \text{ with answer "yes"} \right\}.$$

Solving the problem amounts to recognizing $L$.

**Example (even-parity check; simplest real-world analogue).**    Given a binary string (e.g., a received message), the decision question is: *does it have an even number of 1s?* Over $\Sigma = \{0, 1\}$ define

$$L_{\text{EVEN1}} = \left\{ x \in \{0, 1\}^* : \#_1(x) \text{ is even} \right\},$$

where $\#_1(x)$ counts the 1s in $x$. Membership in $L_{\text{EVEN1}}$ exactly captures passing an *even-parity* check, a standard error-detection rule in data transmission.

## B    DETAILS OF DATASET

We restate some details of the languages included in the FLaRe benchmark from Butoi et al. (2025), licensed under `CC BY 4.0`. It includes datasets for 18 languages from different levels of the Chomsky hierarchy, including regular, deterministic context-free, context-free, and context-sensitive languages. For each language, there is a train, two validation, and one test set. There are two validation sets that can be used for analyzing either the neural network's inductive bias or expressivity (see Butoi et al. (2025) for details). To test the neural networks' length generalization abilities, each test set includes strings that are much longer than those in the train set (maximum length 40 for the train set vs. maximum length 500 for the test set).

For each of the 18 languages, we subsample 100 out of the 10 thousand examples from the train set, ensuring that we sample 50 positive and 50 negative examples. Similarly, we subsample 100 out of the 5010 test examples, ensuring that we have an equal number of positive and negative examples.

We sample negative strings without conditioning on provenance (adversarial vs. random) and keep the class balance fixed. Consequently, our 100-example training and test subsets preserve FLaRe's original mixture: the proportion of adversarially constructed to randomly generated examples matches that of the corresponding FLaRe splits.

Table 4: Formal languages included in the FLaRe benchmark (Butoi et al., 2025). For each language, we include a description, an example, and the language class from the Chomsky hierarchy to which it belongs: regular **(R)**, deterministic context-free **(DCF)**, context-free **(CF)**, or context-sensitive **(CS)**. Let $c_{\boldsymbol{u}}(\boldsymbol{w})$ be the number of times substring $\boldsymbol{u}$ occurs in $\boldsymbol{w}$, let $\boldsymbol{w}^R$ be $\boldsymbol{w}$ reversed, let $\boldsymbol{w}_{i \to a}$ be $\boldsymbol{w}$ with its $i^{\text{th}}$ symbol replaced with $a$, and let $\langle x \rangle$ be the little-endian binary encoding of $x \in \mathbb{Z}_{\geq 0}$.

| Class | Language | Description | Example String |
|---|---|---|---|
| **R** | Even Pairs | $\{\boldsymbol{w} \in \{0,1\}^* \mid c_{01}(\boldsymbol{w}) + c_{10}(\boldsymbol{w}) \text{ is even}\}$ $= \{a\boldsymbol{u}a \mid a \in \{0,1\}, \boldsymbol{u} \in \{0,1\}^*\} \cup \{\varepsilon, 0, 1\}$ | `010110` |
| | Repeat 01 | $\{(01)^n \mid n \geq 0\}$ | `010101` |
| | Parity | $\{\boldsymbol{w} \in \{0,1\}^* \mid c_1(\boldsymbol{w}) \text{ is odd}\}$ | `11011001` |
| | Cycle Navigation | A sequence of left ($<$), right ($>$), stay ($=$) moves on a 5-position cycle, then the final position (0-indexed). | `>>=<>2` |
| | Modular Arithmetic | Expression involving $\{+, -, \times\}$ and $\{0, \ldots, 4\}$, then the result mod 5. No operator precedence. | `1-3×2=1` |
| | Dyck-$(2,3)$ | Strings of balanced brackets with 2 bracket types and a maximum depth of 3. | `[()([])]()` |
| | First | $\{1\boldsymbol{w} \mid \boldsymbol{w} \in \{0,1\}^*\}$ | `100010` |
| **DCF** | Majority | $\{\boldsymbol{w} \in \{0,1\}^* \mid c_1(\boldsymbol{w}) > c_0(\boldsymbol{w})\}$ | `101101` |
| | Stack Manipulation | A stack from bottom to top, a sequence of push and pop operations, and the resulting stack from top to bottom. | `011 POP =10` |
| | Marked Reversal | $\{\boldsymbol{w}\#\boldsymbol{w}^R \mid \boldsymbol{w} \in \{0,1\}^*\}$ | `001#100` |
| **CF** | Unmarked Reversal | $\{\boldsymbol{w}\boldsymbol{w}^R \mid \boldsymbol{w} \in \{0,1\}^*\}$ | `001100` |
| **CS** | Marked Copy | $\{\boldsymbol{w}\#\boldsymbol{w} \mid \boldsymbol{w} \in \{0,1\}^*\}$ | `001#001` |
| | Missing Duplicate | $\{(\boldsymbol{w}\boldsymbol{w})_{i \to \_} \mid \boldsymbol{w} \in \{0,1\}^*, 1 \leq i \leq 2|\boldsymbol{w}|, (\boldsymbol{w}\boldsymbol{w})_i = 1\}$ | `1_011101` |
| | Odds First | $\{a_1 b_1 \cdots a_n b_n a \# a_1 \cdots a_n a b_1 \cdots b_n \mid$ $n \geq 0; a_i, b_i \in \{0,1\}; a \in \{0,1,\varepsilon\}\}$ | `01010=00011` |
| | Binary Addition | $\{\langle x \rangle 0^i + \langle y \rangle 0^j = \langle x+y \rangle 0^k \mid x, y, i, j, k \in \mathbb{Z}_{\geq 0}\}$ | `110+01=10100` |
| | Binary Multiplication | $\{\langle x \rangle 0^i \times \langle y \rangle 0^j = \langle xy \rangle 0^k \mid x, y, i, j, k \in \mathbb{Z}_{\geq 0}\}$ | `110×0100=011` |
| | Compute Sqrt | $\{\langle x \rangle 0^i = \langle \lfloor \sqrt{x} \rfloor \rangle 0^j \mid x, i, j \in \mathbb{Z}_{\geq 0}\}$ | `01010=1100` |
| | Bucket Sort | Sequence of integers in $\{1, \ldots, 5\}$, then # and the sorted sequence. | `45134#13445` |

# C EXPERIMENTAL DETAILS

This appendix provides additional information about the experimental setup, including details about the transformer model trained from scratch and the cost of running the experiments.

# D EXPERIMENTAL DETAILS

## D.1 MODEL ARCHITECTURE

We implement the transformer model trained from scratch using PyTorch's `nn.Transformer`, following the original architecture (Vaswani et al., 2017). It is a causally masked decoder-only

model comprising five layers and approximately $64\,000$ parameters. Input tokens are projected into a $d$-dimensional embedding space and augmented with sinusoidal positional encodings, and each token attends only to preceding tokens via masked self-attention, mirroring the autoregressive behavior of the pretrained LLMs. Each transformer block consists of an 8-head self-attention module followed by a feed-forward network with $4d$ hidden units. We employ residual connections with pre-norm layer normalization before each sublayer (and an additional norm after the final block), and we apply a uniform dropout rate throughout—on embeddings, attention weights, and feed-forward activations—as in the standard implementation. Every sequence is prefixed with a BOS token. We train the model in a supervised manner on a 90/10 train–validation split of our data and evaluate on the same held-out test set used in the ICL experiments.

For all experiments involving DeepSeek-V3 and GPT-4o mini, we use their respective public APIs.

We also evaluate four instruction-tuned checkpoints from Hugging Face, invoked via `vLLM` with transformers tokenizers and each model's official chat template (`AutoTokenizer.apply_chat_template`). Decoding uses `temperature=0.2`, `top_p=1.0`, `n=1`, `seed=42`. We cap context at `max_model_len=8192` and use `dtype=float16`. We stop on EOS and, when present, Llama's `<|eot_id|>`. For zero-shot reasoning prompt we allow up to `max_tokens=1000`; for immediate output prompt we return a single token (`max_tokens=1`).

| Model | HF repository | Params |
|---|---|---|
| Qwen2.5 7B | `Qwen/Qwen2.5-7B-Instruct` | $\sim$7B |
| Qwen2.5 32B *Instruct* | `Qwen/Qwen2.5-32B-Instruct` | $\sim$32B |
| Llama-3.1 8B *Instruct* | `meta-llama/Llama-3.1-8B-Instruct` | $\sim$8B |
| Llama-3.1 70B *Instruct* | `meta-llama/Llama-3.1-70B-Instruct` | $\sim$70B |

Table 5: Instruction-tuned open models evaluated.

## D.2 COST

The cost of running inference on large language models (LLMs) via commercial APIs is typically billed on a per-token basis, with separate rates for input (prompt) and output (completion) tokens. In our experiments, we evaluate DeepSeek, which applies differentiated input-token pricing depending on cache hits versus cache misses, and ChatGPT-4o-mini operating in batch mode. Table 6 summarizes the per-million-token rates for both input and output for each model.

Table 6: API inference costs for 1 M input/output tokens (costs reported as of 15 May 2025).

| Model | Token Type | Cost per 1M Tokens ($) |
|---|---|---|
| DeepSeek-V3 | Input (cache hit) | 0.07 |
| | Input (cache miss) | 0.27 |
| | Output | 1.10 |
| GPT-4o mini (batch) | Input | 0.55 |
| | Output | 2.20 |

Running the full evaluation pipeline via API calls incurred costs of approximately \$50 for DeepSeek-V3 and \$30 for GPT-4o mini. These figures cover all prompting strategies, encoding configurations, and test runs.

## E    EXAMPLES OF PROMPTS AND RESPONSES

We show examples of prompts and outputs from ChatGPT-4o used in our immediate output (App. E.1) and zero-shot reasoning experiments (App. E.2).

## E.1  IMMEDIATE OUTPUT

**Input:**

Here are some positive and negative examples of strings in a formal language. Each example is on a separate line and follows the format "label: string", where label is 0 or 1 (meaning negative or positive, respectively), and string is a sequence of symbols separated by spaces. A blank line indicates the end of the examples.
0: + 0 0 1 0 0 0 = 1 = + 0 + 1 0 0 1 + 0 + 0 = = + 0 = 1 = 0 + = = = + + = + 0 1 +
0: 1 1 1 0 1 0 1 0 1 0 1 =
1: 1 + 1 0 0 0 0 0 = 0 1 0
1: 1 0 0 1 0 0 0 0 + 0 = 1 0 0 1
1: 1 1 + 0 1 1 0 1 1 1 0 0 1 1 0 0 0 0 0 0 0 0 = 1 0 0 1 1 1 1 0 0 1 1 0
0: 1 0 1 + = 0 0 =
0: 0 0 0 1 1 0 1 1 + 0 1 0 0 1 0 0 0 0 0 0 + 0 = 0 1 0 1 0 1 1 1
0: 1 + 0 0 + 1 1
0: 0 1 0 0 1 1 + 0 1 0 1 1 0 0 0 0 1 1 0 0 1 0 0 0
1: 0 1 1 1 + 0 0 1 0 0 1 0 1 0 0 0 0 0 0 = 0 1 0 0 1 1 0 1 0
1: 1 1 0 + 0 0 0 1 1 0 0 0 0 0 0 0 0 = 1 1 0 1 1
0: = + 0 1 0 + +
1: 0 1 0 1 1 1 1 1 1 0 + 0 0 1 1 0 = 0 1 1 0 0 0 0 0 0 1 0 0 0 0 0 0 0 0 0 0
1: 0 1 + 0 = 0 1 0 0 0 0
0: 0 = = 1 0 = 1 + + = = = 1 = + + 1 1 + = 0 1
0: + + 0 0 0 = 1 1 1 + + 0 = = = + 0 = + 0 1 = 1 1 + + + = 0 =
0: 1 0 0 + + 1 0 = 0 0 1 0
1: 0 1 1 0 + 0 0 1 0 1 0 0 0 1 0 0 0 = 0 1 0 1 1 0 0 0 1
1: 1 1 0 0 0 + 0 = 1 1
1: 1 1 1 1 + 1 0 0 0 1 1 1 0 1 1 0 0 0 0 0 0 0 0 0 0 0 = 0 0 0 0 0 0 0 1 1 1
0: + 0 1 0 0 1 0 0 + 0 1 1 0 0 1 0 0 0 0 0 0 0 0 0 0 0 0 0 = 1 1 0 1 0 1 0 0 0
1: 0 0 0 0 0 0 0 0 0 0 0 0 0 0 + 1 1 0 0 = 1 1 0
1: 1 1 1 0 + 0 0 = 1 1 1
1: 1 0 0 0 0 0 0 + 0 1 = 1 1
1: 0 1 1 0 1 0 1 0 1 + 1 0 1 0 0 1 0 0 0 0 1 0 = 1 1 0 1 1 1 1 0 1 0 1 0 0
1: 0 0 1 1 1 1 + 1 1 0 0 0 0 0 = 1 1 1 1 1 1
1: 1 1 0 0 0 0 0 0 1 1 0 0 0 1 0 0 0 + 0 = 1 1 0 0 0 0 0 0 1 1 0 0 0 1 0
0: 0 = 0 1 1 0 = 1 1 + 0 = = 0 0
1: 0 0 0 + 1 0 1 0 1 0 0 0 0 = 1 0 1 0 1
0: 0 0 0 0 0 0 0 0 1 0 + 1 0 0 0 0 0 0 0 0 0 0 = 0 1
1: 0 1 0 + 0 1 0 0 = 0 0 1 0 0
0: 1 = 1 0 0 0 0 1 1 0 1 + 1 = 0 1 + = 0 0 0 = 1 = + + 0 1 = + = = = = =
0: = 0 = 0 1 1 = 1 = 1 + 1 + + + = 0 1 = = 0
1: 1 1 1 0 1 + 1 0 0 0 1 1 1 1 0 1 0 1 0 0 0 0 = 0 0 0 1 0 0 0 0 1 1 0 1
1: 1 0 1 1 0 0 0 0 0 0 + 1 0 0 0 0 0 0 0 0 0 0 0 0 0 0 0 0 = 0 1 1 1
1: 0 0 + 1 = 1 0
1: 0 0 1 + 0 = 0 0 1
0: 1 0 1 1 0 0 = 0
0: 0 1 + + = 1 = = 0 + 0 + 0 + 0 0 = 0 0
1: 1 + 1 0 1 = 0 1 1 0 0 0 0 0 0 0 0 0 0 0 0
0: 0
1: 1 1 0 0 1 0 + 1 = 0 0 1 0 1
0: + = 1 = 1
1: 0 + 0 = 0 0
1: 0 1 0 0 1 1 + 0 1 = 0 0 1 0 1 1 0 0 0
1: 1 0 + 0 0 0 0 0 0 0 = 1 0 0 0
0: + 1 + 0 + 1 1 = 0 + 1 + + 0 = 0 0 + + + + 1 1 = 1 + 0 1 0 0 + 0 1
0: 1 0 0 + 1 = 0 1 0 0 0 = 0 1 1 1
0: = 0 0 0 = =
0: 1 0 + 1 1 = 1 1

```
0: = = 1 = + 1 1 1 + 1 0 0 1 = 0 = 0 = = 1 0 = +
0: 1 + 0 + = + + 1 0 0 = 0
1: 0 + 1 0 0 0 0 0 0 0 = 1 0
0: + 0 = 1 1 = + 0 0 = 0 = 0 + + 0 + = + 1 0 = 1 1 + + +
0: + 0 0 0 + 1 = 0 1 0 0 0 0
0: + 1 0 = 1
0: 1 = 0 0 + = 0 = + 1 0 = 1 + + 1 0 = = 1 +
0: 1 + 0 0 0 0 = 1 1
0: + 1 = 1 0 + = 1 1 = = + 1 +
0: + = = 0 1 1 + = + + + 1 1 0 + = = = = 0 0 = = = + = + = = = +
1: 0 1 0 0 0 0 0 0 0 0 0 0 0 0 0 0 0 0 0 0 0 0 + 1 0 0 0 0 0 0 0 0 0 0 = 1 1
0: 0 0 0 1 1 0 0 0 0 0 0 0 0 0 0 + 0 0 1 0 0 0 0 0 0 0 0 0 0 0 0 = = 0 1 1 1
0: 1 = 1 + 0 + + = = 0 = + = 0 0 + = + 1 1 =
0: 1 = + 0
1: 0 0 1 0 0 0 0 0 0 0 0 0 0 0 0 0 0 + 0 1 0 1 1 1 0 0 0 0 0 = 0 1 1 1 1 1
0:
0: 0 = = 0 1 = 1 0 0
0: = 0 + = = 1 = 1 = 1
0: 1 1 1 1 1 0 + 1 0 1 0 = 0 = 1 0 0 1 0 0 0 0 0 0 0 0 0
1: 0 0 1 0 0 0 + 1 1 1 0 1 0 0 0 0 0 0 0 0 0 0 0 0 0 0 0 0 0 0 0 0 0 = 1 1 0 1 1
0:
0: 0 + 1 = 0
0: 0 1 0 1 1 0 1 + + 0 0 = 0 0 1 1 0 1 0 0 0 0 0
1: 1 + 1 = 0 1 0 0 0
0: = = 0 + = = = + = + + 0 1 = = 1 0 + + + = 1 = 0 0 = = 0
1: 1 1 1 0 0 0 0 0 0 0 0 0 + 1 1 0 0 0 1 = 0 1 0 1 0 1 0
1: 0 0 0 1 1 + 1 1 0 = 1 1 0 1 1
1: 1 0 0 0 0 0 0 0 + 0 1 1 = 1 1 1
0: 0 1 0 0 0 0 0 0 0 0 0 0 = 0 0 0 0 0 0 0 0 0 0 0 + 1 0 0 0 0 0 0 0 = 1 1
1: 1 1 1 0 0 0 1 0 0 0 0 0 0 0 + 1 1 1 1 1 0 0 0 = 0 1 1 0 0 1 1
1: 1 0 0 0 0 0 0 0 0 0 0 + 0 0 0 0 1 1 0 0 0 0 0 = 1 0 0 0 1 1
1: 0 0 + 0 0 0 0 = 0 0
0: 1 0 1 1 1 0 1 1 0 0 0 0 + 0 0 1 0 1 0 0 0 0 0 0 0 0 0 0 = 1 0 0 0 1 1 + 1 1
1: 1 1 0 + 0 0 1 1 0 1 1 1 0 = 1 1 1 1 0 1 1 1
1: 1 + 0 = 1
0: 0 0 0 1 0 0 1 + 1 0 0 = 1 0 1 1 + = 1 0 0 0 1 0 + + 1 +
1: 1 0 + 1 1 1 = 0 0 0 1 0 0 0 0 0 0 0 0 0 0 0 0 0 0 0 0 0 0 0 0 0 0 0 0 0 0
1: 0 0 0 0 1 1 0 0 0 + 1 1 1 1 0 0 0 0 0 0 = 1 1 1 1 1 1
1: 1 1 + 1 1 0 1 0 0 0 = 0 1 1 1
1: 1 1 1 1 1 1 1 1 0 0 + 0 0 0 0 0 0 0 0 0 0 0 0 0 = 1 1 1 1 1 1 1 1
1: 0 0 1 1 1 0 + 0 = 0 0 1 1 1 0 0 0 0 0 0 0 0 0 0 0 0 0 0
1: 1 0 + 1 1 0 0 1 0 1 0 1 = 0 0 1 0 1 0 1 0 1 0 0 0 0 0
0: = = 1 + + 0 = = + 0 0 + 1 = 1 0 0 = 1 = 1 + 1 + + 1 1 + 1 1 0 0 = 0 1 = +
0: 1 1 1 0 = 1 1 0 0 1 0 =
1: 0 0 0 0 + 1 1 = 1 1
1: 0 0 1 0 1 0 0 0 0 0 0 + 1 1 0 1 0 0 0 0 0 0 0 0 0 0 = 1 1 1 1 1
0: = 0 1 = 0 0 0 1 + 1 1 0 0 1 = = 1
1: 0 0 0 0 0 0 + 1 1 1 1 = 1 1 1 1
1: 0 1 + 1 0 1 0 1 = 1 1 1 0 1 0 0 0 0 0 0 0 0 0 0 0 0
0: 1 1 0 0 1 1 0 0 0 0 0 0 0 0 + 0 0 0 0 0 + 1 1 = 0 1 1 0 1 1
```

Given these examples, predict the label (either 0 or 1) of the string after the text "string: " on the following line. If there is no text after "string: ", it represents the empty string. Do not provide any explanation, but respond only with the label 0 or 1. Do not use special markup for the label; just use the text 0 or 1.
string: 0 1 0 1 0 + 1 1 0 0 1 1 0 0 0 1 1 0 = 1 0 1 1 1 1 0 0 0 1 1 0 0 0 0 0 0 0 0 0 0

**Output:**

1

## E.2 ZERO-SHOT REASONING

**Input:**

Here are some positive and negative examples of strings in a formal language. Each example is on a separate line and follows the format "label: string", where label is 0 or 1 (meaning negative or positive, respectively), and string is a sequence of symbols separated by spaces. A blank line indicates the end of the examples.

```
0: + 0 0 1 0 0 0 = 1 = + 0 + 1 0 0 1 + 0 + 0 = = + 0 = 1 = 0 + = = = + + = + 0 1
+
0: 1 1 1 0 1 0 1 0 1 0 1 =
1: 1 + 1 0 0 0 0 0 = 0 1 0
1: 1 0 0 1 0 0 0 0 + 0 = 1 0 0 1
1: 1 1 + 0 1 1 0 1 1 0 0 1 1 0 0 0 0 0 0 0 0 = 1 0 0 1 1 1 1 0 0 1 1 0
0: 1 0 1 + = 0 0 =
0: 0 0 0 1 1 0 1 1 + 0 1 0 0 1 0 0 0 0 0 0 + 0 = 0 1 0 1 0 1 1 1
0: 1 + 0 0 + 1 1
0: 0 1 0 0 1 1 + 0 1 0 1 1 0 0 0 0 1 1 0 0 1 0 0 0
1: 0 1 1 1 + 0 0 1 0 0 1 0 1 0 0 0 0 0 0 = 0 1 0 0 1 1 0 1 0
1: 1 1 0 + 0 0 0 1 1 0 0 0 0 0 0 0 0 = 1 1 0 1 1
0: = + 0 1 0 + +
1: 0 1 0 1 1 1 1 1 1 0 + 0 0 1 1 0 = 0 1 1 0 0 0 0 0 0 1 0 0 0 0 0 0 0 0 0 0
1: 0 1 + 0 = 0 1 0 0 0 0
0: 0 = = 1 0 = 1 + + = = = 1 = + + 1 1 + = 0 1
0: + + 0 0 0 = 1 1 1 + + 0 = = = + 0 = + 0 1 = 1 1 + + + = 0 =
0: 1 0 0 + + 1 0 = 0 0 1 0
1: 0 1 1 0 + 0 0 1 0 1 0 0 0 1 0 0 0 = 0 1 0 1 1 0 0 0 1
1: 1 1 0 0 0 + 0 = 1 1
1: 1 1 1 1 + 1 0 0 0 1 1 1 0 1 1 0 0 0 0 0 0 0 0 0 0 0 = 0 0 0 0 0 0 0 1 1 1
0: + 0 1 0 0 1 0 0 + 0 1 1 0 0 1 0 0 0 0 0 0 0 0 0 0 0 0 0 = 1 1 0 1 0 1 0 0 0
1: 0 0 0 0 0 0 0 0 0 0 0 0 0 0 + 1 1 0 0 = 1 1 0
1: 1 1 1 0 + 0 0 = 1 1 1
1: 1 0 0 0 0 0 0 + 0 1 = 1 1
1: 0 1 1 0 1 0 1 0 1 + 1 0 1 0 0 1 0 0 0 0 1 0 = 1 1 0 1 1 1 1 0 1 0 1 0 0
1: 0 0 1 1 1 1 + 1 1 0 0 0 0 0 = 1 1 1 1 1 1
1: 1 1 0 0 0 0 0 0 1 1 0 0 0 1 0 0 0 + 0 = 1 1 0 0 0 0 0 0 1 1 0 0 0 1 0
0: 0 = 0 1 1 0 = 1 1 + 0 = = 0 0
1: 0 0 0 + 1 0 1 0 1 0 0 0 0 = 1 0 1 0 1
0: 0 0 0 0 0 0 0 0 1 0 + 1 0 0 0 0 0 0 0 0 0 0 = 0 1
1: 0 1 0 + 0 1 0 0 = 0 0 1 0 0
0: 1 = 1 0 0 0 0 1 1 0 1 + 1 = 0 1 + = 0 0 0 = 1 = + + 0 1 = + = = = = = =
0: = 0 = 0 1 1 = 1 = 1 + 1 + + + = 0 1 = = 0
1: 1 1 1 0 1 + 1 0 0 0 1 1 1 1 0 1 0 1 0 0 0 0 = 0 0 0 1 0 0 0 0 1 1 0 1
1: 1 0 1 1 0 0 0 0 0 0 + 1 0 0 0 0 0 0 0 0 0 0 0 0 0 0 0 0 0 = 0 1 1 1
1: 0 0 + 1 = 1 0
1: 0 0 1 + 0 = 0 0 1
0: 1 0 1 1 0 0 = 0
0: 0 1 + + = 1 = = 0 + 0 + 0 + 0 0 = 0 0
1: 1 + 1 0 1 = 0 1 1 0 0 0 0 0 0 0 0 0 0 0
0: 0
1: 1 1 0 0 1 0 + 1 = 0 0 1 0 1
0: + = 1 = 1
1: 0 + 0 = 0 0
```

```
1: 0 1 0 0 1 1 + 0 1 = 0 0 1 0 1 1 0 0 0
1: 1 0 + 0 0 0 0 0 0 0 0 = 1 0 0 0
0: + 1 + 0 + 1 1 = 0 + 1 + + 0 = 0 0 + + + + 1 1 = 1 + 0 1 0 0 + 0 1
0: 1 0 0 + 1 = 0 1 0 0 0 = 0 1 1 1
0: = 0 0 0 = =
0: 1 0 + 1 1 = 1 1
0: = = 1 = + 1 1 1 + 1 0 0 1 = 0 = 0 = = 1 0 = +
0: 1 + 0 + = + + 1 0 0 = 0 0
1: 0 + 1 0 0 0 0 0 0 0 = 1 0
0: + 0 = 1 1 = + 0 0 = 0 = 0 + + 0 + = + 1 0 = 1 1 + + +
0: + 0 0 0 + 1 = 0 1 0 0 0 0
0: + 1 0 = 1
0: 1 = 0 0 + = 0 = + 1 0 = 1 + + 1 0 = = 1 +
0: 1 + 0 0 0 0 = 1 1
0: + 1 = 1 0 + = 1 1 = = + 1 +
0: + = = 0 1 1 + = + + + 1 1 0 + = = = = 0 0 = = = + = + = = = +
1: 0 1 0 0 0 0 0 0 0 0 0 0 0 0 0 0 0 0 0 0 0 + 1 0 0 0 0 0 0 0 0 0 0 = 1 1
0: 0 0 0 1 1 0 0 0 0 0 0 0 0 0 0 + 0 0 1 0 0 0 0 0 0 0 0 0 0 0 0 = = 0 1 1 1
0: 1 = 1 + 0 + + = = 0 = + = 0 0 + = + 1 1 =
0: 1 = + 0
1: 0 0 1 0 0 0 0 0 0 0 0 0 0 0 0 0 0 + 0 1 0 1 1 1 0 0 0 0 0 = 0 1 1 1 1 1
0:
0: 0 = = 0 1 = 1 0 0
0: = 0 + = = 1 = 1 = 1
0: 1 1 1 1 1 0 + 1 0 1 0 = 0 = 1 0 0 1 0 0 0 0 0 0 0 0 0
1: 0 0 1 0 0 0 + 1 1 1 0 1 0 0 0 0 0 0 0 0 0 0 0 0 0 0 0 0 0 0 = 1 1 0 1 1
0:
0: 0 + 1 = 0
0: 0 1 0 1 1 0 1 + + 0 0 = 0 0 1 1 0 1 0 0 0 0 0
1: 1 + 1 = 0 1 0 0 0
0: = = 0 + = = = + = + + 0 1 = = 1 0 + + + = 1 = 0 0 = = 0
1: 1 1 1 0 0 0 0 0 0 0 0 0 + 1 1 0 0 0 1 = 0 1 0 1 0 1 0
1: 0 0 0 1 1 + 1 1 0 = 1 1 0 1 1
1: 1 0 0 0 0 0 0 0 + 0 1 1 = 1 1 1
0: 0 1 0 0 0 0 0 0 0 0 0 = 0 0 0 0 0 0 0 0 0 0 0 + 1 0 0 0 0 0 0 0 = 1 1
1: 1 1 1 0 0 0 1 0 0 0 0 0 0 0 + 1 1 1 1 1 0 0 0 = 0 1 1 0 0 1 1
1: 1 0 0 0 0 0 0 0 0 0 0 + 0 0 0 0 1 1 0 0 0 0 0 = 1 0 0 0 1 1
1: 0 0 + 0 0 0 0 = 0 0
0: 1 0 1 1 1 0 1 1 0 0 0 0 + 0 0 1 0 1 0 0 0 0 0 0 0 0 0 = 1 0 0 0 1 1 + 1 1
1: 1 1 0 + 0 0 1 1 0 1 1 1 0 = 1 1 1 1 0 1 1 1
1: 1 + 0 = 1
0: 0 0 0 1 0 0 1 + 1 0 0 = 1 0 1 1 + = 1 0 0 0 1 0 + + 1 +
1: 1 0 + 1 1 1 = 0 0 0 1 0 0 0 0 0 0 0 0 0 0 0 0 0 0 0 0 0 0 0 0 0 0 0 0
1: 0 0 0 0 1 1 0 0 0 + 1 1 1 1 0 0 0 0 0 0 = 1 1 1 1 1 1
1: 1 1 + 1 1 0 1 0 0 0 = 0 1 1 1
1: 1 1 1 1 1 1 1 1 0 0 + 0 0 0 0 0 0 0 0 0 0 0 0 0 = 1 1 1 1 1 1 1 1
1: 0 0 1 1 1 0 + 0 = 0 0 1 1 1 0 0 0 0 0 0 0 0 0 0 0 0 0 0
1: 1 0 + 1 1 0 0 1 0 1 0 1 = 0 0 1 0 1 0 1 0 1 0 0 0 0 0
0: = = 1 + + 0 = = + 0 0 + 1 = 1 0 0 = 1 = 1 + 1 + + 1 1 + 1 1 0 0 = 0 1 = +
0: 1 1 1 0 = 1 1 0 0 1 0 =
1: 0 0 0 0 + 1 1 = 1 1
1: 0 0 1 0 1 0 0 0 0 0 0 + 1 1 0 1 0 0 0 0 0 0 0 0 0 = 1 1 1 1 1
0: = 0 1 = 0 0 0 1 + 1 1 0 0 1 = = 1
1: 0 0 0 0 0 0 + 1 1 1 1 = 1 1 1 1
1: 0 1 + 1 0 1 0 1 = 1 1 1 0 1 0 0 0 0 0 0 0 0 0 0 0 0
0: 1 1 0 0 1 1 0 0 0 0 0 0 0 0 + 0 0 0 0 0 + 1 1 = 0 1 1 0 1 1
```

Given these examples, predict the label (either 0 or 1) of the string after the text "string: " on the following line. If there is no text after "string: ", it represents the empty string. Feel free to think step-by-step about the problem, but at the end of your response, respond only with the label 0 or 1. Do not use special markup for the label; just use the text 0 or 1.
string: 0 1 0 1 0 + 1 1 0 0 1 1 0 0 0 1 1 0 = 1 0 1 1 1 1 0 0 0 1 1 0 0 0 0 0 0 0 0 0 0

**Output:**

To determine the label (0 or 1) for the given string, let's analyze the patterns observed in the provided examples:

Observations:
Presence of = and +:

Many positive examples (label 1) contain = or +, but their presence alone is not sufficient to determine the label, as some negative examples (label 0) also include these symbols.

The key seems to be the context or positioning of these symbols within the string.

Structure of Positive Examples:

Positive examples often have = or + appearing in a way that seems to divide or structure the string meaningfully. For example:

1 + 1 0 0 0 0 0 = 0 1 0 (label 1)

0 1 + 0 = 0 1 0 0 0 0 (label 1)

The symbols = or + are not overly repeated or clustered in a way that seems arbitrary.

Structure of Negative Examples:

Negative examples often have excessive or clustered repetitions of = or +, or these symbols appear in a way that seems random or unbalanced.
For example:

+ 0 0 1 0 0 0 = 1 = + 0 + 1 0 0 1 + 0 + 0 = = + 0 = 1 = 0 + = = = + + = + 0 1 + (label 0)

= = 1 = + 1 1 1 + 1 0 0 1 = 0 = 0 = = 1 0 = + (label 0)

Given String Analysis:

The string is: 0 1 0 1 0 + 1 1 0 0 1 1 0 0 0 1 1 0 = 1 0 1 1 1 1 0 0 0 1 1 0 0 0 0 0 0 0 0 0 0

It contains one + and one =, neither of which is excessively repeated or clustered.

The + and = appear to divide the string into meaningful segments, similar to positive examples.

The structure does not resemble the chaotic or repetitive patterns seen in negative examples.

Conclusion:

> The given string follows the pattern of positive examples (label 1), where = and + are used meaningfully without excessive repetition or randomness.
>
> 1

# F  PROMPT TUNING

Before running the large-scale sweep, we ran a small prompt-tuning pilot on two languages—DYCK-2-3 and MAJORITY—using a single DeepSeek model. We crossed six prompt templates (`prompt1`–`prompt6`) with several encoding strategies: many→one, one→one, and one→many with fan-out $k \in \{2, 3, 4, 5\}$. Results are reported in Table 7.

Overall, prompt choice had only a modest effect. Within a fixed encoding, scores varied little across prompts: for DYCK-2-3, the within-encoding spread was typically $\leq 0.10$ (e.g., many→one varied by $0.04$; one→one by $0.10$; $k$=2 by $0.02$). For MAJORITY, many→one, one→one, and $k$=2 each varied by $0.04$ across prompts. The larger differences came from the encoding itself, not the prompt. In short, prompt tuning did not change much, and we therefore proceeded to the full experiments without additional per-task prompt optimization. All prompt templates are available in our repository.

| Language | Prompting Strategy | Encoding Strategy | | | | | |
| | | many → one | one → one | one → many | | | |
| | | | | 2 | 3 | 4 | 5 |
| --- | --- | --- | --- | --- | --- | --- | --- |
| dyck-2-3 | prompt1 | 0.72 | 0.66 | 0.70 | 0.64 | 0.68 | 0.68 |
| | prompt2 | 0.72 | 0.76 | 0.70 | 0.62 | 0.66 | 0.64 |
| | prompt3 | 0.72 | 0.70 | 0.70 | 0.66 | 0.66 | 0.62 |
| | prompt4 | 0.70 | 0.68 | 0.70 | 0.64 | 0.66 | 0.72 |
| | prompt5 | 0.68 | 0.72 | 0.72 | 0.62 | 0.62 | 0.60 |
| | prompt6 | 0.70 | 0.70 | 0.70 | 0.64 | 0.64 | 0.62 |
| majority | prompt1 | 0.88 | 0.84 | 0.86 | 0.66 | 0.74 | 0.70 |
| | prompt2 | 0.84 | 0.86 | 0.86 | 0.56 | 0.74 | 0.68 |
| | prompt3 | 0.86 | 0.86 | 0.84 | 0.58 | 0.74 | 0.72 |
| | prompt4 | 0.86 | 0.86 | 0.84 | 0.78 | 0.62 | 0.68 |
| | prompt5 | 0.86 | 0.82 | 0.86 | 0.60 | 0.64 | 0.70 |
| | prompt6 | 0.84 | 0.86 | 0.82 | 0.58 | 0.54 | 0.66 |

Table 7: Prompt-tuning pilot on two languages (DYCK-2-3, MAJORITY) with a single DeepSeek model. We cross six prompt templates (`prompt1`–`prompt6`) with encoding strategies: many→one, one→one, and one→many with fan-out $k \in 2, 3, 4, 5$; cells report accuracy. Prompt choice had only modest effect (typically $\leq 0.10$ within a fixed encoding); performance varied more by encoding (many→one and one→one strongest; one→many degrading with larger $k$), so we did not pursue further prompt tuning before the large-scale experiments.

# G  EXTENDED RESULTS

Tables report test accuracy on the held-out set for each formal language, broken down by model, prompt template, and encoding strategy. Boldface marks the best per language (across all models/prompts/encodings); blue marks the best per (language, model) (across prompts/encodings); if both apply, bold wins.

## G.1  ALL ACCURACY RESULTS (BY LANGUAGE × MODEL × PROMPT × ENCODING)

Table 8, Table 9, and Table 10 report test accuracy on the held-out set, organized by language class (R; DCF+CF; CS). Each table breaks results down by model, prompt template (immediate output

and zero-shot reasoning), and encoding strategy (many→one, one→one, and variants 2–5). Bold indicates the best score per language (across all models/prompts/encodings), and blue indicates the best score per (language, model) (across prompts/encodings). The tables are intended to let readers: (i) identify, for each language, which prompt/encoding yields the top score overall and per model, and (ii) compare models under the same prompt/encoding. All values are shown exactly as produced.

## G.2 ALL ERROR-RATE RESULTS (BY LANGUAGE × MODEL × PROMPT × ENCODING)

Table 11, Table 12, and Table 13 report the percentage of invalid generations (i.e., unparsable outputs), organized by language class (R; DCF+CF; CS). Each table breaks results down by model, prompt template (immediate output and zero-shot reasoning), and encoding strategy (many → one, one → one, and one → many with $t \in \{2, 3, 4, 5\}$). Across models and languages, many → one and one → one are almost always 0% error. In contrast, larger encodings (especially variants 3–5) exhibit the highest error rates, with the most frequent spikes for Qwen2.5 32B: e.g., R-class languages show $22 - 99\%$ on enc. 3 and up to 97% on enc. 5; in DCF, *stack-manipulation* reaches 99% (IO, enc. 3); in CS, *binary-addition* hits 98% (IO, one → many with $t=3$) and *bucket-sort* reaches $74 - 94\%$ under ZSR (enc. 3–4). A single Llama-3.1 8B outlier attains 100% errors (*bucket-sort*, IO, enc. 5). These patterns are consistent with longer/heavier encodings increasing prompt length and structural overhead, which likely raises the risk of token-budget pressure or instruction confusion, yielding unparsable outputs. Practically, to minimize invalid generations, prefer compact encodings (many→one / one→one) and avoid late-index encodings (especially $\geq 3$) for Qwen2.5 32B and similar settings. All values are shown exactly as produced. Notably, the proprietary LLMs prompted via APIs (GPT-4o mini and DeepSeek-V3) never produced an unparsable response.

## G.3 ACCURACY VS. STRING LENGTH

For each formal language, we first select the single best ICL configuration—the maximum test accuracy across all models, prompting templates, and encoding techniques (ties broken arbitrarily). We then re-evaluate that configuration at the level of individual test examples and associate each example with its input length, measured as the number of symbols in the original string (i.e., independent of the encoding scheme). Lengths are bucketed into contiguous bins of width $20$ ($1 - 20$, $21 - 40$, ...) up to the longest string observed, and accuracy within a bin is the fraction of examples correctly classified in that bin (examples where the model produced an unparsable output are counted as incorrect). Bins with no test cases are left empty. Each panel therefore shows how performance varies with string length for the *best* setup for that language; captions under each panel report the model and configuration (prompt template and encoding technique) used for that plot.

Across languages, we do not observe a consistent length-dependent pattern. In particular, accuracy does not uniformly degrade with longer strings; some languages show flat or non-monotonic trends, while others exhibit task-specific (or dataset specific) fluctuations. Overall, input length alone does not explain success or failure in our setting as seen in Figure 3

| Class | Language | Model | Prompting Strategy | Encoding Strategy | | | | | |
|---|---|---|---|---|---|---|---|---|---|
| | | | | many → one | one → one | 2 | 3 | 4 | 5 |
| R | even-pairs | GPT-4o mini | immediate output | 0.59 | 0.46 | 0.41 | 0.38 | 0.41 | 0.40 |
| | | | zero-shot reasoning | 0.58 | 0.57 | 0.54 | 0.47 | 0.42 | 0.42 |
| | | DeepSeek-V3 | immediate output | 0.47 | 0.49 | 0.52 | 0.49 | 0.56 | 0.59 |
| | | | zero-shot reasoning | 0.50 | 0.52 | 0.56 | 0.50 | 0.51 | 0.55 |
| | | Qwen2.5 7B | immediate output | 0.54 | 0.49 | 0.49 | 0.50 | 0.50 | 0.50 |
| | | | zero-shot reasoning | 0.54 | 0.52 | 0.47 | 0.50 | 0.50 | 0.50 |
| | | Qwen2.5 32B | immediate output | 0.55 | 0.56 | 0.58 | 0.48 | 0.28 | 0.12 |
| | | | zero-shot reasoning | 0.56 | 0.55 | 0.54 | 0.56 | 0.52 | 0.50 |
| | | Llama-3.1-70B-Instruct | immediate output | 0.54 | 0.50 | 0.56 | 0.51 | 0.51 | **0.61** |
| | | | zero-shot reasoning | 0.51 | 0.50 | 0.52 | 0.54 | 0.55 | 0.52 |
| | | Llama-3.1 8B | immediate output | 0.50 | 0.50 | 0.50 | 0.50 | 0.50 | 0.50 |
| | | | zero-shot reasoning | 0.48 | 0.50 | 0.53 | 0.50 | 0.50 | 0.50 |
| | repeat-01 | GPT-4o mini | immediate output | 0.56 | 0.77 | 0.69 | 0.64 | 0.62 | 0.62 |
| | | | zero-shot reasoning | 0.55 | 0.79 | 0.75 | 0.70 | 0.65 | 0.61 |
| | | DeepSeek-V3 | immediate output | **0.97** | 0.84 | 0.82 | 0.74 | 0.75 | 0.71 |
| | | | zero-shot reasoning | 0.96 | 0.95 | 0.88 | 0.77 | 0.77 | 0.73 |
| | | Qwen2.5 7B | immediate output | 0.60 | 0.54 | 0.51 | 0.50 | 0.50 | 0.51 |
| | | | zero-shot reasoning | 0.60 | 0.62 | 0.50 | 0.50 | 0.50 | 0.50 |
| | | Qwen2.5 32B | immediate output | 0.74 | 0.61 | 0.73 | 0.71 | 0.31 | 0.20 |
| | | | zero-shot reasoning | 0.74 | 0.74 | 0.71 | 0.67 | 0.73 | 0.71 |
| | | Llama-3.1-70B-Instruct | immediate output | 0.77 | 0.67 | 0.55 | 0.61 | 0.56 | 0.50 |
| | | | zero-shot reasoning | 0.67 | 0.56 | 0.50 | 0.51 | 0.53 | 0.56 |
| | | Llama-3.1 8B | immediate output | 0.50 | 0.50 | 0.50 | 0.50 | 0.52 | 0.50 |
| | | | zero-shot reasoning | 0.54 | 0.51 | 0.50 | 0.50 | 0.50 | 0.50 |
| | parity | GPT-4o mini | immediate output | 0.60 | 0.52 | 0.53 | 0.51 | 0.50 | 0.49 |
| | | | zero-shot reasoning | **0.66** | 0.50 | 0.53 | 0.46 | 0.45 | 0.46 |
| | | DeepSeek-V3 | immediate output | 0.58 | 0.50 | 0.48 | 0.49 | 0.45 | 0.52 |
| | | | zero-shot reasoning | 0.49 | 0.46 | 0.45 | 0.54 | 0.39 | 0.45 |
| | | Qwen2.5 7B | immediate output | 0.51 | 0.50 | 0.50 | 0.51 | 0.51 | 0.51 |
| | | | zero-shot reasoning | 0.48 | 0.51 | 0.51 | 0.51 | 0.51 | 0.51 |
| | | Qwen2.5 32B | immediate output | 0.48 | 0.48 | 0.49 | 0.52 | 0.22 | 0.10 |
| | | | zero-shot reasoning | 0.48 | 0.51 | 0.48 | 0.50 | 0.48 | 0.57 |
| | | Llama-3.1-70B-Instruct | immediate output | 0.47 | 0.46 | 0.49 | 0.49 | 0.50 | 0.50 |
| | | | zero-shot reasoning | 0.48 | 0.50 | 0.50 | 0.50 | 0.49 | 0.50 |
| | | Llama-3.1 8B | immediate output | 0.49 | 0.50 | 0.50 | 0.50 | 0.50 | 0.50 |
| | | | zero-shot reasoning | 0.51 | 0.50 | 0.50 | 0.50 | 0.51 | 0.50 |
| | cycle-navigation | GPT-4o mini | immediate output | 0.73 | 0.80 | 0.80 | 0.82 | 0.82 | 0.79 |
| | | | zero-shot reasoning | 0.73 | **0.84** | 0.73 | 0.83 | 0.79 | 0.75 |
| | | DeepSeek-V3 | immediate output | 0.69 | 0.71 | 0.79 | 0.82 | 0.82 | 0.82 |
| | | | zero-shot reasoning | 0.69 | 0.77 | 0.79 | 0.78 | 0.79 | 0.83 |
| | | Qwen2.5 7B | immediate output | 0.79 | 0.50 | 0.65 | 0.55 | 0.59 | 0.60 |
| | | | zero-shot reasoning | 0.79 | 0.50 | 0.51 | 0.55 | 0.58 | 0.61 |
| | | Qwen2.5 32B | immediate output | 0.83 | 0.82 | 0.82 | 0.62 | 0.18 | 0.02 |
| | | | zero-shot reasoning | 0.81 | 0.82 | 0.76 | 0.77 | 0.53 | 0.54 |
| | | Llama-3.1-70B-Instruct | immediate output | 0.83 | 0.82 | 0.82 | 0.82 | 0.82 | 0.82 |
| | | | zero-shot reasoning | 0.82 | 0.68 | 0.81 | 0.54 | 0.80 | 0.80 |
| | | Llama-3.1 8B | immediate output | 0.50 | 0.50 | 0.50 | 0.50 | 0.50 | 0.50 |
| | | | zero-shot reasoning | 0.56 | 0.50 | 0.50 | 0.50 | 0.50 | 0.50 |
| | modular-arithmetic-simple | GPT-4o mini | immediate output | 0.51 | 0.68 | 0.60 | 0.62 | 0.56 | 0.60 |
| | | | zero-shot reasoning | 0.48 | 0.55 | 0.58 | 0.61 | 0.51 | 0.53 |
| | | DeepSeek-V3 | immediate output | 0.75 | 0.75 | 0.70 | 0.55 | 0.56 | 0.53 |
| | | | zero-shot reasoning | 0.54 | 0.49 | 0.52 | 0.59 | 0.52 | 0.53 |
| | | Qwen2.5 7B | immediate output | 0.63 | 0.49 | 0.55 | 0.50 | 0.61 | 0.54 |
| | | | zero-shot reasoning | 0.63 | 0.53 | 0.54 | 0.52 | 0.56 | 0.55 |
| | | Qwen2.5 32B | immediate output | 0.77 | 0.76 | 0.71 | 0.62 | 0.62 | 0.71 |
| | | | zero-shot reasoning | **0.79** | 0.72 | 0.70 | 0.74 | 0.70 | 0.62 |
| | | Llama-3.1-70B-Instruct | immediate output | 0.73 | 0.51 | 0.56 | 0.53 | 0.50 | 0.52 |
| | | | zero-shot reasoning | 0.54 | 0.51 | 0.53 | 0.54 | 0.50 | 0.51 |
| | | Llama-3.1 8B | immediate output | 0.50 | 0.50 | 0.52 | 0.50 | 0.50 | 0.50 |
| | | | zero-shot reasoning | 0.52 | 0.51 | 0.50 | 0.50 | 0.50 | 0.50 |
| | dyck-2-3 | GPT-4o mini | immediate output | 0.60 | 0.65 | 0.63 | 0.64 | 0.60 | 0.57 |
| | | | zero-shot reasoning | 0.56 | 0.61 | 0.72 | 0.62 | 0.59 | 0.52 |
| | | DeepSeek-V3 | immediate output | 0.68 | 0.67 | 0.63 | 0.51 | 0.50 | 0.51 |
| | | | zero-shot reasoning | 0.66 | 0.52 | 0.53 | 0.65 | 0.66 | 0.63 |
| | | Qwen2.5 7B | immediate output | 0.72 | 0.53 | 0.57 | 0.53 | 0.53 | 0.59 |
| | | | zero-shot reasoning | 0.69 | 0.59 | 0.59 | 0.55 | 0.49 | 0.50 |
| | | Qwen2.5 32B | immediate output | **0.76** | 0.57 | 0.67 | 0.01 | 0.62 | 0.55 |
| | | | zero-shot reasoning | 0.75 | 0.57 | 0.38 | 0.51 | 0.58 | 0.42 |
| | | Llama-3.1-70B-Instruct | immediate output | 0.73 | 0.52 | 0.55 | 0.64 | 0.56 | 0.57 |
| | | | zero-shot reasoning | 0.56 | 0.57 | 0.50 | 0.55 | 0.58 | 0.55 |
| | | Llama-3.1 8B | immediate output | 0.50 | 0.50 | 0.50 | 0.49 | 0.50 | 0.52 |
| | | | zero-shot reasoning | 0.45 | 0.53 | 0.51 | 0.50 | 0.50 | 0.50 |
| | first | GPT-4o mini | immediate output | 0.60 | 0.99 | 1.00 | 1.00 | 1.00 | 1.00 |
| | | | zero-shot reasoning | 0.61 | 1.00 | 1.00 | 0.94 | 0.48 | 0.45 |
| | | DeepSeek-V3 | immediate output | 0.92 | 1.00 | 1.00 | 1.00 | 1.00 | 1.00 |
| | | | zero-shot reasoning | 0.97 | 0.96 | 0.98 | 1.00 | 0.99 | 1.00 |
| | | Qwen2.5 7B | immediate output | 0.68 | 0.50 | 0.50 | 0.50 | 0.50 | 0.50 |
| | | | zero-shot reasoning | 0.76 | 0.50 | 0.50 | 0.50 | 0.50 | 0.51 |
| | | Qwen2.5 32B | immediate output | 0.99 | 0.92 | 0.96 | 0.98 | 0.65 | 0.16 |
| | | | zero-shot reasoning | 1.00 | 0.98 | 1.00 | 0.98 | 0.99 | 0.97 |
| | | Llama-3.1-70B-Instruct | immediate output | 0.98 | 0.99 | 0.98 | 1.00 | 1.00 | 0.99 |
| | | | zero-shot reasoning | 0.97 | 0.58 | 1.00 | 0.78 | 1.00 | 0.94 |
| | | Llama-3.1 8B | immediate output | 0.50 | 0.50 | 0.50 | 0.50 | 0.50 | 0.52 |
| | | | zero-shot reasoning | 0.52 | 0.51 | 0.50 | 0.50 | 0.50 | 0.50 |

Table 8: Test accuracy on the held-out set for Class R languages, broken down by model, prompt template, and encoding strategy. Bold marks the best per language; blue marks the best per (language, model).

| Class | Language | Model | Prompting Strategy | Encoding Strategy | | | | | |
| | | | | many → one | one → one | 2 | 3 | 4 | 5 |
|---|---|---|---|---|---|---|---|---|---|
| DCF | majority | GPT-4o mini | immediate output | 0.79 | 0.78 | 0.77 | 0.75 | 0.67 | 0.64 |
| | | | zero-shot reasoning | 0.77 | 0.75 | 0.76 | 0.79 | 0.57 | 0.51 |
| | | DeepSeek-V3 | immediate output | 0.81 | 0.85 | 0.76 | 0.81 | 0.77 | 0.70 |
| | | | zero-shot reasoning | 0.79 | 0.78 | 0.80 | 0.79 | 0.70 | 0.73 |
| | | Qwen2.5 7B | immediate output | 0.58 | 0.55 | 0.53 | 0.52 | 0.52 | 0.52 |
| | | | zero-shot reasoning | 0.67 | 0.52 | 0.52 | 0.52 | 0.52 | 0.54 |
| | | Qwen2.5 32B | immediate output | 0.87 | 0.76 | 0.86 | 0.77 | 0.48 | 0.11 |
| | | | zero-shot reasoning | 0.88 | 0.73 | 0.84 | 0.70 | 0.68 | 0.68 |
| | | Llama-3.1-70B-Instruct | immediate output | 0.74 | 0.59 | 0.56 | 0.64 | 0.63 | 0.55 |
| | | | zero-shot reasoning | 0.67 | 0.55 | 0.68 | 0.52 | 0.58 | 0.59 |
| | | Llama-3.1 8B | immediate output | 0.50 | 0.50 | 0.50 | 0.50 | 0.33 | 0.52 |
| | | | zero-shot reasoning | 0.52 | 0.53 | 0.51 | 0.52 | 0.50 | 0.50 |
| | stack-manipulation | GPT-4o mini | immediate output | 0.45 | 0.68 | 0.70 | 0.57 | 0.67 | 0.61 |
| | | | zero-shot reasoning | 0.46 | 0.72 | 0.69 | 0.62 | 0.54 | 0.54 |
| | | DeepSeek-V3 | immediate output | 0.72 | 0.76 | 0.75 | 0.69 | 0.66 | 0.60 |
| | | | zero-shot reasoning | 0.74 | 0.74 | 0.64 | 0.71 | 0.71 | 0.68 |
| | | Qwen2.5 7B | immediate output | 0.75 | 0.51 | 0.66 | 0.48 | 0.51 | 0.54 |
| | | | zero-shot reasoning | 0.74 | 0.56 | 0.65 | 0.59 | 0.55 | 0.62 |
| | | Qwen2.5 32B | immediate output | 0.75 | 0.76 | 0.75 | 0.01 | 0.34 | 0.61 |
| | | | zero-shot reasoning | 0.75 | 0.76 | 0.73 | 0.74 | 0.68 | -1.00 |
| | | Llama-3.1-70B-Instruct | immediate output | 0.74 | 0.72 | 0.59 | 0.64 | 0.51 | 0.63 |
| | | | zero-shot reasoning | 0.72 | 0.54 | 0.70 | 0.64 | 0.50 | 0.63 |
| | | Llama-3.1 8B | immediate output | 0.50 | 0.50 | 0.50 | 0.34 | 0.50 | 0.50 |
| | | | zero-shot reasoning | 0.51 | 0.50 | 0.50 | 0.50 | 0.50 | 0.50 |
| | marked-reversal | GPT-4o mini | immediate output | 0.56 | 0.68 | 0.65 | 0.64 | 0.62 | 0.63 |
| | | | zero-shot reasoning | 0.62 | 0.67 | 0.67 | 0.69 | 0.65 | 0.62 |
| | | DeepSeek-V3 | immediate output | 0.69 | 0.68 | 0.69 | 0.62 | 0.68 | 0.68 |
| | | | zero-shot reasoning | 0.62 | 0.64 | 0.67 | 0.69 | 0.63 | 0.68 |
| | | Qwen2.5 7B | immediate output | 0.56 | 0.51 | 0.52 | 0.40 | 0.55 | 0.47 |
| | | | zero-shot reasoning | 0.55 | 0.52 | 0.55 | 0.54 | 0.51 | 0.58 |
| | | Qwen2.5 32B | immediate output | 0.69 | 0.67 | 0.63 | 0.28 | 0.50 | 0.57 |
| | | | zero-shot reasoning | 0.70 | 0.69 | 0.68 | 0.68 | 0.69 | 0.69 |
| | | Llama-3.1-70B-Instruct | immediate output | 0.63 | 0.57 | 0.67 | 0.66 | 0.66 | 0.67 |
| | | | zero-shot reasoning | 0.50 | 0.51 | 0.52 | 0.56 | 0.64 | 0.59 |
| | | Llama-3.1 8B | immediate output | 0.50 | 0.50 | 0.50 | 0.50 | 0.50 | 0.50 |
| | | | zero-shot reasoning | 0.51 | 0.50 | 0.50 | 0.49 | 0.50 | 0.50 |
| CF | unmarked-reversal | GPT-4o mini | immediate output | 0.50 | 0.46 | 0.46 | 0.46 | 0.46 | 0.46 |
| | | | zero-shot reasoning | 0.53 | 0.47 | 0.47 | 0.48 | 0.46 | 0.45 |
| | | DeepSeek-V3 | immediate output | 0.53 | 0.46 | 0.48 | 0.47 | 0.50 | 0.45 |
| | | | zero-shot reasoning | 0.54 | 0.50 | 0.48 | 0.47 | 0.50 | 0.53 |
| | | Qwen2.5 7B | immediate output | 0.49 | 0.55 | 0.51 | 0.50 | 0.50 | 0.50 |
| | | | zero-shot reasoning | 0.52 | 0.53 | 0.51 | 0.49 | 0.51 | 0.50 |
| | | Qwen2.5 32B | immediate output | 0.50 | 0.49 | 0.51 | 0.44 | 0.39 | 0.11 |
| | | | zero-shot reasoning | 0.52 | 0.48 | 0.47 | 0.50 | 0.51 | |
| | | Llama-3.1-70B-Instruct | immediate output | 0.51 | 0.53 | 0.53 | 0.51 | 0.50 | 0.50 |
| | | | zero-shot reasoning | 0.50 | 0.50 | 0.50 | 0.52 | 0.49 | 0.50 |
| | | Llama-3.1 8B | immediate output | 0.50 | 0.50 | 0.50 | 0.50 | 0.50 | 0.50 |
| | | | zero-shot reasoning | 0.52 | 0.55 | 0.49 | 0.50 | 0.50 | 0.50 |

Table 9: Test accuracy on the held-out set for Classes DCF and CF languages, broken down by model, prompt template, and encoding strategy. Bold marks the best per language; blue marks the best per (language, model).

| Class | Language | Model | Prompting Strategy | Encoding Strategy | | | | | |
|---|---|---|---|---|---|---|---|---|---|
| | | | | many → one | one → one | 2 | 3 | 4 | 5 |
| CS | marked-copy | GPT-4o mini | immediate output | 0.69 | 0.70 | 0.64 | 0.64 | 0.56 | 0.60 |
| | | | zero-shot reasoning | 0.66 | 0.71 | 0.69 | 0.67 | 0.54 | 0.55 |
| | | DeepSeek-V3 | immediate output | 0.76 | 0.73 | 0.75 | 0.69 | 0.70 | 0.77 |
| | | | zero-shot reasoning | 0.79 | 0.68 | 0.64 | 0.71 | 0.71 | 0.74 |
| | | Qwen2.5 7B | immediate output | 0.57 | 0.48 | 0.51 | 0.39 | 0.61 | 0.53 |
| | | | zero-shot reasoning | 0.56 | 0.49 | 0.62 | 0.58 | 0.55 | 0.64 |
| | | Qwen2.5 32B | immediate output | 0.76 | 0.69 | 0.58 | 0.19 | 0.48 | 0.46 |
| | | | zero-shot reasoning | 0.76 | 0.76 | 0.75 | 0.77 | 0.78 | 0.75 |
| | | Llama-3.1-70B-Instruct | immediate output | 0.60 | 0.51 | 0.64 | 0.74 | 0.62 | 0.67 |
| | | | zero-shot reasoning | 0.51 | 0.56 | 0.54 | 0.53 | 0.50 | 0.50 |
| | | Llama-3.1 8B | immediate output | 0.50 | 0.50 | 0.48 | 0.50 | 0.50 | 0.52 |
| | | | zero-shot reasoning | 0.44 | 0.50 | 0.48 | 0.50 | 0.50 | 0.50 |
| | missing-duplicate-string | GPT-4o mini | immediate output | 0.52 | 0.73 | 0.63 | 0.63 | 0.53 | 0.46 |
| | | | zero-shot reasoning | 0.51 | 0.69 | 0.64 | 0.68 | 0.58 | 0.56 |
| | | DeepSeek-V3 | immediate output | 0.69 | 0.76 | 0.72 | 0.62 | 0.68 | 0.67 |
| | | | zero-shot reasoning | 0.73 | 0.64 | 0.71 | 0.71 | 0.74 | 0.64 |
| | | Qwen2.5 7B | immediate output | 0.76 | 0.50 | 0.48 | 0.50 | 0.44 | 0.50 |
| | | | zero-shot reasoning | 0.74 | 0.49 | 0.50 | 0.50 | 0.50 | 0.50 |
| | | Qwen2.5 32B | immediate output | 0.71 | 0.59 | 0.56 | 0.59 | 0.49 | 0.15 |
| | | | zero-shot reasoning | 0.74 | 0.71 | 0.56 | 0.63 | 0.65 | 0.53 |
| | | Llama-3.1-70B-Instruct | immediate output | 0.54 | 0.72 | 0.72 | 0.68 | 0.67 | 0.67 |
| | | | zero-shot reasoning | 0.50 | 0.63 | 0.75 | 0.73 | 0.59 | 0.68 |
| | | Llama-3.1 8B | immediate output | 0.50 | 0.50 | 0.50 | 0.50 | 0.50 | 0.51 |
| | | | zero-shot reasoning | 0.50 | 0.49 | 0.51 | 0.49 | 0.49 | 0.50 |
| | odds-first | GPT-4o mini | immediate output | 0.63 | 0.67 | 0.65 | 0.63 | 0.62 | 0.60 |
| | | | zero-shot reasoning | 0.46 | 0.67 | 0.65 | 0.69 | 0.60 | 0.61 |
| | | DeepSeek-V3 | immediate output | 0.66 | 0.68 | 0.66 | 0.66 | 0.68 | 0.68 |
| | | | zero-shot reasoning | 0.75 | 0.71 | 0.60 | 0.56 | 0.68 | 0.64 |
| | | Qwen2.5 7B | immediate output | 0.55 | 0.51 | 0.55 | 0.45 | 0.54 | 0.48 |
| | | | zero-shot reasoning | 0.54 | 0.61 | 0.57 | 0.63 | 0.63 | 0.66 |
| | | Qwen2.5 32B | immediate output | 0.62 | 0.64 | 0.61 | 0.22 | 0.55 | 0.51 |
| | | | zero-shot reasoning | 0.68 | 0.67 | 0.68 | 0.69 | 0.69 | 0.68 |
| | | Llama-3.1-70B-Instruct | immediate output | 0.67 | 0.63 | 0.67 | 0.69 | 0.68 | 0.68 |
| | | | zero-shot reasoning | 0.59 | 0.62 | 0.67 | 0.67 | 0.62 | 0.64 |
| | | Llama-3.1 8B | immediate output | 0.50 | 0.50 | 0.52 | 0.51 | 0.50 | 0.50 |
| | | | zero-shot reasoning | 0.47 | 0.51 | 0.50 | 0.50 | 0.51 | 0.50 |
| | binary-addition | GPT-4o mini | immediate output | 0.58 | 0.76 | 0.79 | 0.77 | 0.72 | 0.73 |
| | | | zero-shot reasoning | 0.61 | 0.67 | 0.74 | 0.80 | 0.68 | 0.66 |
| | | DeepSeek-V3 | immediate output | 0.76 | 0.76 | 0.74 | 0.73 | 0.67 | 0.69 |
| | | | zero-shot reasoning | 0.67 | 0.78 | 0.75 | 0.80 | 0.77 | 0.80 |
| | | Qwen2.5 7B | immediate output | 0.67 | 0.53 | 0.51 | 0.52 | 0.67 | 0.61 |
| | | | zero-shot reasoning | 0.69 | 0.60 | 0.54 | 0.60 | 0.66 | 0.64 |
| | | Qwen2.5 32B | immediate output | 0.75 | 0.77 | 0.77 | 0.02 | 0.62 | 0.62 |
| | | | zero-shot reasoning | 0.75 | 0.77 | 0.44 | 0.62 | 0.66 | 0.37 |
| | | Llama-3.1-70B-Instruct | immediate output | 0.75 | 0.74 | 0.76 | 0.73 | 0.71 | 0.74 |
| | | | zero-shot reasoning | 0.61 | 0.70 | 0.73 | 0.77 | 0.54 | 0.68 |
| | | Llama-3.1 8B | immediate output | 0.50 | 0.50 | 0.50 | 0.52 | 0.50 | 0.50 |
| | | | zero-shot reasoning | 0.53 | 0.50 | 0.51 | 0.50 | 0.50 | 0.50 |
| | binary-multiplication | GPT-4o mini | immediate output | 0.49 | 0.67 | 0.76 | 0.69 | 0.54 | 0.57 |
| | | | zero-shot reasoning | 0.63 | 0.68 | 0.72 | 0.76 | 0.70 | 0.66 |
| | | DeepSeek-V3 | immediate output | 0.72 | 0.60 | 0.76 | 0.77 | 0.76 | 0.71 |
| | | | zero-shot reasoning | 0.73 | 0.63 | 0.66 | 0.74 | 0.71 | 0.59 |
| | | Qwen2.5 7B | immediate output | 0.72 | 0.50 | 0.54 | 0.50 | 0.52 | 0.60 |
| | | | zero-shot reasoning | 0.72 | 0.52 | 0.53 | 0.62 | 0.61 | 0.62 |
| | | Qwen2.5 32B | immediate output | 0.76 | 0.77 | 0.76 | 0.10 | 0.52 | 0.41 |
| | | | zero-shot reasoning | 0.76 | 0.76 | 0.34 | 0.62 | 0.41 | 0.29 |
| | | Llama-3.1-70B-Instruct | immediate output | 0.76 | 0.76 | 0.76 | 0.77 | 0.76 | 0.75 |
| | | | zero-shot reasoning | 0.76 | 0.66 | 0.72 | 0.72 | 0.73 | 0.72 |
| | | Llama-3.1 8B | immediate output | 0.50 | 0.50 | 0.50 | 0.50 | 0.50 | 0.50 |
| | | | zero-shot reasoning | 0.62 | 0.52 | 0.48 | 0.50 | 0.50 | 0.50 |
| | compute-sqrt | GPT-4o mini | immediate output | 0.51 | 0.60 | 0.74 | 0.73 | 0.68 | 0.66 |
| | | | zero-shot reasoning | 0.52 | 0.61 | 0.68 | 0.70 | 0.67 | 0.65 |
| | | DeepSeek-V3 | immediate output | 0.74 | 0.74 | 0.69 | 0.75 | 0.75 | 0.75 |
| | | | zero-shot reasoning | 0.70 | 0.80 | 0.68 | 0.71 | 0.74 | 0.69 |
| | | Qwen2.5 7B | immediate output | 0.72 | 0.50 | 0.60 | 0.42 | 0.53 | 0.54 |
| | | | zero-shot reasoning | 0.71 | 0.58 | 0.67 | 0.52 | 0.61 | 0.50 |
| | | Qwen2.5 32B | immediate output | 0.75 | 0.64 | 0.72 | 0.66 | 0.54 | 0.04 |
| | | | zero-shot reasoning | 0.75 | 0.76 | 0.70 | 0.74 | 0.75 | 0.75 |
| | | Llama-3.1-70B-Instruct | immediate output | 0.70 | 0.75 | 0.75 | 0.71 | 0.66 | 0.71 |
| | | | zero-shot reasoning | 0.63 | 0.72 | 0.73 | 0.72 | 0.72 | 0.72 |
| | | Llama-3.1 8B | immediate output | 0.51 | 0.50 | 0.55 | 0.56 | 0.51 | 0.50 |
| | | | zero-shot reasoning | 0.56 | 0.48 | 0.51 | 0.50 | 0.50 | 0.50 |
| | bucket-sort | GPT-4o mini | immediate output | 0.50 | 0.66 | 0.67 | 0.63 | 0.66 | 0.63 |
| | | | zero-shot reasoning | 0.45 | 0.44 | 0.45 | 0.41 | 0.49 | 0.55 |
| | | DeepSeek-V3 | immediate output | 0.66 | 0.63 | 0.57 | 0.54 | 0.58 | 0.52 |
| | | | zero-shot reasoning | 0.70 | 0.62 | 0.49 | 0.53 | 0.49 | 0.44 |
| | | Qwen2.5 7B | immediate output | 0.64 | 0.50 | 0.54 | 0.50 | 0.50 | 0.50 |
| | | | zero-shot reasoning | 0.65 | 0.45 | 0.67 | 0.54 | 0.51 | 0.49 |
| | | Qwen2.5 32B | immediate output | 0.58 | 0.67 | 0.55 | 0.57 | 0.44 | 0.54 |
| | | | zero-shot reasoning | 0.55 | 0.65 | 0.54 | 0.15 | 0.02 | 0.50 |
| | | Llama-3.1-70B-Instruct | immediate output | 0.58 | 0.67 | 0.52 | 0.50 | 0.52 | 0.50 |
| | | | zero-shot reasoning | 0.51 | 0.55 | 0.52 | 0.50 | 0.50 | 0.50 |
| | | Llama-3.1 8B | immediate output | 0.50 | 0.51 | 0.50 | 0.51 | 0.50 | 0.00 |
| | | | zero-shot reasoning | 0.50 | 0.52 | 0.50 | 0.50 | 0.50 | 0.50 |

Table 10: Test accuracy on the held-out set for Class CS languages, broken down by model, prompt template, and encoding strategy. Bold marks the best per language; blue marks the best per (language, model).

| Class | Language | Model | Prompting Strategy | Encoding Strategy (% with model_output = -1) | | | | | |
|---|---|---|---|---|---|---|---|---|---|
| | | | | many → one | one → one | 2 | 3 | 4 | 5 |
| R | even-pairs | GPT-4o mini | immediate output | 0.0% | 0.0% | 0.0% | 0.0% | 0.0% | 0.0% |
| | | | zero-shot reasoning | 0.0% | 0.0% | 0.0% | 0.0% | 0.0% | 0.0% |
| | | DeepSeek-V3 | immediate output | 0.0% | 0.0% | 0.0% | 0.0% | 0.0% | 0.0% |
| | | | zero-shot reasoning | 0.0% | 0.0% | 0.0% | 0.0% | 0.0% | 0.0% |
| | | Qwen2.5 7B | immediate output | 0.0% | 0.0% | 0.0% | 0.0% | 0.0% | 0.0% |
| | | | zero-shot reasoning | 0.0% | 0.0% | 0.0% | 0.0% | 0.0% | 0.0% |
| | | Qwen2.5 32B | immediate output | 0.0% | 0.0% | 0.0% | 0.0% | 39.0% | 73.0% |
| | | | zero-shot reasoning | 0.0% | 0.0% | 0.0% | 0.0% | 0.0% | 0.0% |
| | | Llama-3.1-70B-Instruct | immediate output | 0.0% | 0.0% | 0.0% | 0.0% | 0.0% | 0.0% |
| | | | zero-shot reasoning | 0.0% | 0.0% | 0.0% | 0.0% | 0.0% | 0.0% |
| | | Llama-3.1 8B | immediate output | 0.0% | 0.0% | 0.0% | 0.0% | 0.0% | 0.0% |
| | | | zero-shot reasoning | 2.0% | 0.0% | 0.0% | 0.0% | 0.0% | 0.0% |
| | repeat-01 | GPT-4o mini | immediate output | 0.0% | 0.0% | 0.0% | 0.0% | 0.0% | 0.0% |
| | | | zero-shot reasoning | 0.0% | 0.0% | 0.0% | 0.0% | 0.0% | 0.0% |
| | | DeepSeek-V3 | immediate output | 0.0% | 0.0% | 0.0% | 0.0% | 0.0% | 0.0% |
| | | | zero-shot reasoning | 0.0% | 0.0% | 0.0% | 0.0% | 0.0% | 0.0% |
| | | Qwen2.5 7B | immediate output | 0.0% | 0.0% | 0.0% | 0.0% | 0.0% | 0.0% |
| | | | zero-shot reasoning | 0.0% | 0.0% | 0.0% | 0.0% | 0.0% | 0.0% |
| | | Qwen2.5 32B | immediate output | 0.0% | 0.0% | 0.0% | 0.0% | 50.0% | 70.0% |
| | | | zero-shot reasoning | 0.0% | 0.0% | 0.0% | 0.0% | 0.0% | 0.0% |
| | | Llama-3.1-70B-Instruct | immediate output | 0.0% | 0.0% | 0.0% | 0.0% | 0.0% | 0.0% |
| | | | zero-shot reasoning | 0.0% | 0.0% | 0.0% | 0.0% | 0.0% | 0.0% |
| | | Llama-3.1 8B | immediate output | 0.0% | 0.0% | 0.0% | 0.0% | 0.0% | 0.0% |
| | | | zero-shot reasoning | 0.0% | 1.0% | 0.0% | 0.0% | 0.0% | 0.0% |
| | parity | GPT-4o mini | immediate output | 0.0% | 0.0% | 0.0% | 0.0% | 0.0% | 0.0% |
| | | | zero-shot reasoning | 0.0% | 0.0% | 0.0% | 0.0% | 0.0% | 0.0% |
| | | DeepSeek-V3 | immediate output | 0.0% | 0.0% | 0.0% | 0.0% | 0.0% | 0.0% |
| | | | zero-shot reasoning | 0.0% | 0.0% | 0.0% | 0.0% | 0.0% | 0.0% |
| | | Qwen2.5 7B | immediate output | 0.0% | 0.0% | 0.0% | 0.0% | 0.0% | 0.0% |
| | | | zero-shot reasoning | 0.0% | 0.0% | 0.0% | 0.0% | 0.0% | 0.0% |
| | | Qwen2.5 32B | immediate output | 0.0% | 0.0% | 0.0% | 0.0% | 53.0% | 76.0% |
| | | | zero-shot reasoning | 0.0% | 0.0% | 0.0% | 0.0% | 0.0% | 0.0% |
| | | Llama-3.1-70B-Instruct | immediate output | 0.0% | 0.0% | 0.0% | 0.0% | 0.0% | 0.0% |
| | | | zero-shot reasoning | 0.0% | 0.0% | 0.0% | 0.0% | 0.0% | 0.0% |
| | | Llama-3.1 8B | immediate output | 0.0% | 0.0% | 0.0% | 0.0% | 0.0% | 0.0% |
| | | | zero-shot reasoning | 0.0% | 0.0% | 0.0% | 0.0% | 0.0% | 0.0% |
| | cycle-navigation | GPT-4o mini | immediate output | 0.0% | 0.0% | 0.0% | 0.0% | 0.0% | 0.0% |
| | | | zero-shot reasoning | 0.0% | 0.0% | 0.0% | 0.0% | 0.0% | 0.0% |
| | | DeepSeek-V3 | immediate output | 0.0% | 0.0% | 0.0% | 0.0% | 0.0% | 0.0% |
| | | | zero-shot reasoning | 0.0% | 0.0% | 0.0% | 0.0% | 0.0% | 0.0% |
| | | Qwen2.5 7B | immediate output | 0.0% | 0.0% | 0.0% | 0.0% | 0.0% | 0.0% |
| | | | zero-shot reasoning | 0.0% | 0.0% | 0.0% | 0.0% | 0.0% | 0.0% |
| | | Qwen2.5 32B | immediate output | 0.0% | 0.0% | 0.0% | 22.0% | 65.0% | 97.0% |
| | | | zero-shot reasoning | 0.0% | 0.0% | 0.0% | 0.0% | 35.0% | 28.0% |
| | | Llama-3.1-70B-Instruct | immediate output | 0.0% | 0.0% | 0.0% | 0.0% | 0.0% | 0.0% |
| | | | zero-shot reasoning | 0.0% | 0.0% | 0.0% | 0.0% | 0.0% | 0.0% |
| | | Llama-3.1 8B | immediate output | 0.0% | 0.0% | 0.0% | 0.0% | 0.0% | 0.0% |
| | | | zero-shot reasoning | 1.0% | 0.0% | 0.0% | 0.0% | 0.0% | 0.0% |
| | modular-arithmetic-simple | GPT-4o mini | immediate output | 0.0% | 0.0% | 0.0% | 0.0% | 0.0% | 0.0% |
| | | | zero-shot reasoning | 0.0% | 0.0% | 0.0% | 0.0% | 0.0% | 0.0% |
| | | DeepSeek-V3 | immediate output | 0.0% | 0.0% | 0.0% | 0.0% | 0.0% | 0.0% |
| | | | zero-shot reasoning | 0.0% | 0.0% | 0.0% | 0.0% | 0.0% | 0.0% |
| | | Qwen2.5 7B | immediate output | 0.0% | 0.0% | 0.0% | 0.0% | 0.0% | 0.0% |
| | | | zero-shot reasoning | 0.0% | 0.0% | 0.0% | 0.0% | 0.0% | 0.0% |
| | | Qwen2.5 32B | immediate output | 0.0% | 0.0% | 0.0% | 0.0% | 0.0% | 0.0% |
| | | | zero-shot reasoning | 0.0% | 0.0% | 0.0% | 0.0% | 0.0% | 0.0% |
| | | Llama-3.1-70B-Instruct | immediate output | 0.0% | 0.0% | 0.0% | 0.0% | 0.0% | 0.0% |
| | | | zero-shot reasoning | 0.0% | 0.0% | 0.0% | 0.0% | 0.0% | 0.0% |
| | | Llama-3.1 8B | immediate output | 0.0% | 0.0% | 0.0% | 0.0% | 0.0% | 0.0% |
| | | | zero-shot reasoning | 0.0% | 0.0% | 0.0% | 0.0% | 0.0% | 0.0% |
| | dyck-2-3 | GPT-4o mini | immediate output | 0.0% | 0.0% | 0.0% | 0.0% | 0.0% | 0.0% |
| | | | zero-shot reasoning | 0.0% | 0.0% | 0.0% | 0.0% | 0.0% | 0.0% |
| | | DeepSeek-V3 | immediate output | 0.0% | 0.0% | 0.0% | 0.0% | 0.0% | 0.0% |
| | | | zero-shot reasoning | 0.0% | 0.0% | 0.0% | 0.0% | 0.0% | 0.0% |
| | | Qwen2.5 7B | immediate output | 0.0% | 0.0% | 0.0% | 0.0% | 0.0% | 0.0% |
| | | | zero-shot reasoning | 0.0% | 0.0% | 0.0% | 0.0% | 0.0% | 0.0% |
| | | Qwen2.5 32B | immediate output | 0.0% | 0.0% | 0.0% | 99.0% | 0.0% | 0.0% |
| | | | zero-shot reasoning | 0.0% | 0.0% | 33.0% | 4.0% | 1.0% | 18.0% |
| | | Llama-3.1-70B-Instruct | immediate output | 0.0% | 0.0% | 0.0% | 0.0% | 0.0% | 0.0% |
| | | | zero-shot reasoning | 0.0% | 0.0% | 0.0% | 0.0% | 0.0% | 0.0% |
| | | Llama-3.1 8B | immediate output | 0.0% | 0.0% | 0.0% | 0.0% | 0.0% | 0.0% |
| | | | zero-shot reasoning | 5.0% | 0.0% | 0.0% | 0.0% | 0.0% | 0.0% |
| | first | GPT-4o mini | immediate output | 0.0% | 0.0% | 0.0% | 0.0% | 0.0% | 0.0% |
| | | | zero-shot reasoning | 0.0% | 0.0% | 0.0% | 0.0% | 0.0% | 0.0% |
| | | DeepSeek-V3 | immediate output | 0.0% | 0.0% | 0.0% | 0.0% | 0.0% | 0.0% |
| | | | zero-shot reasoning | 0.0% | 0.0% | 0.0% | 0.0% | 0.0% | 0.0% |
| | | Qwen2.5 7B | immediate output | 0.0% | 0.0% | 0.0% | 0.0% | 0.0% | 0.0% |
| | | | zero-shot reasoning | 0.0% | 0.0% | 0.0% | 0.0% | 0.0% | 0.0% |
| | | Qwen2.5 32B | immediate output | 0.0% | 0.0% | 0.0% | 0.0% | 35.0% | 84.0% |
| | | | zero-shot reasoning | 0.0% | 0.0% | 0.0% | 0.0% | 0.0% | 0.0% |
| | | Llama-3.1-70B-Instruct | immediate output | 0.0% | 0.0% | 0.0% | 0.0% | 0.0% | 0.0% |
| | | | zero-shot reasoning | 0.0% | 0.0% | 0.0% | 0.0% | 0.0% | 0.0% |
| | | Llama-3.1 8B | immediate output | 0.0% | 0.0% | 0.0% | 0.0% | 0.0% | 0.0% |
| | | | zero-shot reasoning | 0.0% | 0.0% | 0.0% | 0.0% | 0.0% | 0.0% |

Table 11: Percentage of invalid generations for Class R languages, broken down by model, prompt template, and encoding strategy (many→one, one→one, and variants 2–5).

| Class | Language | Model | Prompting Strategy | Encoding Strategy (% with model_output = -1) | | | | | |
|---|---|---|---|---|---|---|---|---|---|
| | | | | many → one | one → one | 2 | 3 | 4 | 5 |
| DCF | majority | GPT-4o mini | immediate output | 0.0% | 0.0% | 0.0% | 0.0% | 0.0% | 0.0% |
| | | | zero-shot reasoning | 0.0% | 0.0% | 0.0% | 0.0% | 0.0% | 0.0% |
| | | DeepSeek-V3 | immediate output | 0.0% | 0.0% | 0.0% | 0.0% | 0.0% | 0.0% |
| | | | zero-shot reasoning | 0.0% | 0.0% | 0.0% | 0.0% | 0.0% | 0.0% |
| | | Qwen2.5 7B | immediate output | 0.0% | 0.0% | 0.0% | 0.0% | 0.0% | 0.0% |
| | | | zero-shot reasoning | 0.0% | 0.0% | 0.0% | 0.0% | 0.0% | 0.0% |
| | | Qwen2.5 32B | immediate output | 0.0% | 0.0% | 0.0% | 0.0% | 29.0% | 79.0% |
| | | | zero-shot reasoning | 0.0% | 0.0% | 0.0% | 0.0% | 0.0% | 0.0% |
| | | Llama-3.1-70B-Instruct | immediate output | 0.0% | 0.0% | 0.0% | 0.0% | 0.0% | 0.0% |
| | | | zero-shot reasoning | 0.0% | 0.0% | 0.0% | 0.0% | 0.0% | 0.0% |
| | | Llama-3.1 8B | immediate output | 0.0% | 0.0% | 0.0% | 0.0% | 45.0% | 0.0% |
| | | | zero-shot reasoning | 3.0% | 0.0% | 0.0% | 0.0% | 0.0% | 0.0% |
| | stack-manipulation | GPT-4o mini | immediate output | 0.0% | 0.0% | 0.0% | 0.0% | 0.0% | 0.0% |
| | | | zero-shot reasoning | 0.0% | 0.0% | 0.0% | 0.0% | 0.0% | 0.0% |
| | | DeepSeek-V3 | immediate output | 0.0% | 0.0% | 0.0% | 0.0% | 0.0% | 0.0% |
| | | | zero-shot reasoning | 0.0% | 0.0% | 0.0% | 0.0% | 0.0% | 0.0% |
| | | Qwen2.5 7B | immediate output | 0.0% | 0.0% | 0.0% | 0.0% | 0.0% | 0.0% |
| | | | zero-shot reasoning | 0.0% | 0.0% | 0.0% | 0.0% | 0.0% | 0.0% |
| | | Qwen2.5 32B | immediate output | 0.0% | 0.0% | 0.0% | 99.0% | 38.0% | 3.0% |
| | | | zero-shot reasoning | 0.0% | 0.0% | 0.0% | 0.0% | 7.0% | |
| | | Llama-3.1-70B-Instruct | immediate output | 0.0% | 0.0% | 0.0% | 0.0% | 0.0% | 0.0% |
| | | | zero-shot reasoning | 0.0% | 0.0% | 0.0% | 0.0% | 0.0% | 0.0% |
| | | Llama-3.1 8B | immediate output | 0.0% | 0.0% | 0.0% | 32.0% | 0.0% | 0.0% |
| | | | zero-shot reasoning | 0.0% | 0.0% | 0.0% | 0.0% | 0.0% | 0.0% |
| | marked-reversal | GPT-4o mini | immediate output | 0.0% | 0.0% | 0.0% | 0.0% | 0.0% | 0.0% |
| | | | zero-shot reasoning | 0.0% | 0.0% | 0.0% | 0.0% | 0.0% | 0.0% |
| | | DeepSeek-V3 | immediate output | 0.0% | 0.0% | 0.0% | 0.0% | 0.0% | 0.0% |
| | | | zero-shot reasoning | 0.0% | 0.0% | 0.0% | 0.0% | 0.0% | 0.0% |
| | | Qwen2.5 7B | immediate output | 0.0% | 0.0% | 0.0% | 0.0% | 0.0% | 0.0% |
| | | | zero-shot reasoning | 0.0% | 0.0% | 0.0% | 0.0% | 0.0% | 0.0% |
| | | Qwen2.5 32B | immediate output | 0.0% | 0.0% | 7.0% | 63.0% | 1.0% | 4.0% |
| | | | zero-shot reasoning | 0.0% | 0.0% | 0.0% | 0.0% | 0.0% | 0.0% |
| | | Llama-3.1-70B-Instruct | immediate output | 0.0% | 0.0% | 0.0% | 0.0% | 0.0% | 0.0% |
| | | | zero-shot reasoning | 0.0% | 0.0% | 0.0% | 0.0% | 0.0% | 0.0% |
| | | Llama-3.1 8B | immediate output | 0.0% | 0.0% | 0.0% | 0.0% | 0.0% | 0.0% |
| | | | zero-shot reasoning | 0.0% | 0.0% | 0.0% | 0.0% | 0.0% | 0.0% |
| CF | unmarked-reversal | GPT-4o mini | immediate output | 0.0% | 0.0% | 0.0% | 0.0% | 0.0% | 0.0% |
| | | | zero-shot reasoning | 0.0% | 0.0% | 0.0% | 0.0% | 0.0% | 0.0% |
| | | DeepSeek-V3 | immediate output | 0.0% | 0.0% | 0.0% | 0.0% | 0.0% | 0.0% |
| | | | zero-shot reasoning | 0.0% | 0.0% | 0.0% | 0.0% | 0.0% | 0.0% |
| | | Qwen2.5 7B | immediate output | 0.0% | 0.0% | 0.0% | 0.0% | 0.0% | 0.0% |
| | | | zero-shot reasoning | 0.0% | 0.0% | 0.0% | 0.0% | 0.0% | 0.0% |
| | | Qwen2.5 32B | immediate output | 0.0% | 0.0% | 0.0% | 0.0% | 27.0% | 80.0% |
| | | | zero-shot reasoning | 0.0% | 0.0% | 0.0% | 0.0% | 0.0% | |
| | | Llama-3.1-70B-Instruct | immediate output | 0.0% | 0.0% | 0.0% | 0.0% | 0.0% | 0.0% |
| | | | zero-shot reasoning | 0.0% | 0.0% | 0.0% | 0.0% | 0.0% | 0.0% |
| | | Llama-3.1 8B | immediate output | 0.0% | 0.0% | 0.0% | 0.0% | 0.0% | 0.0% |
| | | | zero-shot reasoning | 2.0% | 1.0% | 0.0% | 0.0% | 0.0% | 0.0% |

Table 12: Percentage of invalid generations for Classes DCF and CF languages, broken down by model, prompt template, and encoding strategy (many→one, one→one, and variants 2–5).

| Class | Language | Model | Prompting Strategy | Encoding Strategy (% with model_output = -1) | | | | | |
|---|---|---|---|---|---|---|---|---|---|
| | | | | many → one | one → one | 2 | 3 | 4 | 5 |
| CS | marked-copy | GPT-4o mini | immediate output | 0.0% | 0.0% | 0.0% | 0.0% | 0.0% | 0.0% |
| | | | zero-shot reasoning | 0.0% | 0.0% | 0.0% | 0.0% | 0.0% | 0.0% |
| | | DeepSeek-V3 | immediate output | 0.0% | 0.0% | 0.0% | 0.0% | 0.0% | 0.0% |
| | | | zero-shot reasoning | 0.0% | 0.0% | 0.0% | 0.0% | 0.0% | 0.0% |
| | | Qwen2.5 7B | immediate output | 0.0% | 0.0% | 0.0% | 0.0% | 0.0% | 0.0% |
| | | | zero-shot reasoning | 0.0% | 0.0% | 0.0% | 0.0% | 0.0% | 0.0% |
| | | Qwen2.5 32B | immediate output | 0.0% | 0.0% | 6.0% | 69.0% | 2.0% | 7.0% |
| | | | zero-shot reasoning | 0.0% | 0.0% | 0.0% | 0.0% | 0.0% | 0.0% |
| | | Llama-3.1-70B-Instruct | immediate output | 0.0% | 0.0% | 0.0% | 0.0% | 0.0% | 0.0% |
| | | | zero-shot reasoning | 0.0% | 0.0% | 0.0% | 0.0% | 0.0% | 0.0% |
| | | Llama-3.1 8B | immediate output | 0.0% | 0.0% | 0.0% | 0.0% | 0.0% | 0.0% |
| | | | zero-shot reasoning | 1.0% | 0.0% | 0.0% | 0.0% | 0.0% | 0.0% |
| | missing-duplicate-string | GPT-4o mini | immediate output | 0.0% | 0.0% | 0.0% | 0.0% | 0.0% | 0.0% |
| | | | zero-shot reasoning | 0.0% | 0.0% | 0.0% | 0.0% | 0.0% | 0.0% |
| | | DeepSeek-V3 | immediate output | 0.0% | 0.0% | 0.0% | 0.0% | 0.0% | 0.0% |
| | | | zero-shot reasoning | 0.0% | 0.0% | 0.0% | 0.0% | 0.0% | 0.0% |
| | | Qwen2.5 7B | immediate output | 0.0% | 0.0% | 0.0% | 0.0% | 0.0% | 2.0% |
| | | | zero-shot reasoning | 0.0% | 0.0% | 0.0% | 0.0% | 0.0% | 0.0% |
| | | Qwen2.5 32B | immediate output | 0.0% | 0.0% | 0.0% | 5.0% | 2.0% | 71.0% |
| | | | zero-shot reasoning | 0.0% | 0.0% | 0.0% | 0.0% | 0.0% | 0.0% |
| | | Llama-3.1-70B-Instruct | immediate output | 0.0% | 0.0% | 0.0% | 0.0% | 0.0% | 0.0% |
| | | | zero-shot reasoning | 0.0% | 0.0% | 0.0% | 0.0% | 0.0% | 0.0% |
| | | Llama-3.1 8B | immediate output | 0.0% | 0.0% | 0.0% | 0.0% | 0.0% | 0.0% |
| | | | zero-shot reasoning | 0.0% | 0.0% | 0.0% | 1.0% | 0.0% | 0.0% |
| | odds-first | GPT-4o mini | immediate output | 0.0% | 0.0% | 0.0% | 0.0% | 0.0% | 0.0% |
| | | | zero-shot reasoning | 0.0% | 0.0% | 0.0% | 0.0% | 0.0% | 0.0% |
| | | DeepSeek-V3 | immediate output | 0.0% | 0.0% | 0.0% | 0.0% | 0.0% | 0.0% |
| | | | zero-shot reasoning | 0.0% | 0.0% | 0.0% | 0.0% | 0.0% | 0.0% |
| | | Qwen2.5 7B | immediate output | 0.0% | 0.0% | 0.0% | 0.0% | 0.0% | 0.0% |
| | | | zero-shot reasoning | 0.0% | 0.0% | 0.0% | 0.0% | 0.0% | 0.0% |
| | | Qwen2.5 32B | immediate output | 0.0% | 0.0% | 6.0% | 64.0% | 0.0% | 2.0% |
| | | | zero-shot reasoning | 0.0% | 0.0% | 0.0% | 0.0% | 0.0% | 0.0% |
| | | Llama-3.1-70B-Instruct | immediate output | 0.0% | 0.0% | 0.0% | 0.0% | 0.0% | 0.0% |
| | | | zero-shot reasoning | 0.0% | 0.0% | 0.0% | 0.0% | 0.0% | 0.0% |
| | | Llama-3.1 8B | immediate output | 0.0% | 0.0% | 0.0% | 0.0% | 0.0% | 0.0% |
| | | | zero-shot reasoning | 3.0% | 0.0% | 0.0% | 0.0% | 0.0% | 0.0% |
| | binary-addition | GPT-4o mini | immediate output | 0.0% | 0.0% | 0.0% | 0.0% | 0.0% | 0.0% |
| | | | zero-shot reasoning | 0.0% | 0.0% | 0.0% | 0.0% | 0.0% | 0.0% |
| | | DeepSeek-V3 | immediate output | 0.0% | 0.0% | 0.0% | 0.0% | 0.0% | 0.0% |
| | | | zero-shot reasoning | 0.0% | 0.0% | 0.0% | 0.0% | 0.0% | 0.0% |
| | | Qwen2.5 7B | immediate output | 0.0% | 0.0% | 0.0% | 0.0% | 0.0% | 0.0% |
| | | | zero-shot reasoning | 0.0% | 0.0% | 0.0% | 0.0% | 0.0% | 0.0% |
| | | Qwen2.5 32B | immediate output | 0.0% | 0.0% | 0.0% | 98.0% | 0.0% | 0.0% |
| | | | zero-shot reasoning | 0.0% | 0.0% | 33.0% | 8.0% | 3.0% | 33.0% |
| | | Llama-3.1-70B-Instruct | immediate output | 0.0% | 0.0% | 0.0% | 0.0% | 0.0% | 0.0% |
| | | | zero-shot reasoning | 0.0% | 0.0% | 0.0% | 0.0% | 0.0% | 0.0% |
| | | Llama-3.1 8B | immediate output | 0.0% | 0.0% | 0.0% | 0.0% | 0.0% | 0.0% |
| | | | zero-shot reasoning | 7.0% | 0.0% | 0.0% | 0.0% | 0.0% | 0.0% |
| | binary-multiplication | GPT-4o mini | immediate output | 0.0% | 0.0% | 0.0% | 0.0% | 0.0% | 0.0% |
| | | | zero-shot reasoning | 0.0% | 0.0% | 0.0% | 0.0% | 0.0% | 0.0% |
| | | DeepSeek-V3 | immediate output | 0.0% | 0.0% | 0.0% | 0.0% | 0.0% | 0.0% |
| | | | zero-shot reasoning | 0.0% | 0.0% | 0.0% | 0.0% | 0.0% | 0.0% |
| | | Qwen2.5 7B | immediate output | 0.0% | 0.0% | 0.0% | 0.0% | 0.0% | 0.0% |
| | | | zero-shot reasoning | 0.0% | 0.0% | 0.0% | 0.0% | 0.0% | 0.0% |
| | | Qwen2.5 32B | immediate output | 0.0% | 0.0% | 0.0% | 88.0% | 0.0% | 16.0% |
| | | | zero-shot reasoning | 0.0% | 0.0% | 29.0% | 9.0% | 27.0% | 46.0% |
| | | Llama-3.1-70B-Instruct | immediate output | 0.0% | 0.0% | 0.0% | 0.0% | 0.0% | 0.0% |
| | | | zero-shot reasoning | 0.0% | 0.0% | 0.0% | 0.0% | 0.0% | 0.0% |
| | | Llama-3.1 8B | immediate output | 0.0% | 0.0% | 0.0% | 0.0% | 0.0% | 0.0% |
| | | | zero-shot reasoning | 0.0% | 0.0% | 0.0% | 0.0% | 0.0% | 0.0% |
| | compute-sqrt | GPT-4o mini | immediate output | 0.0% | 0.0% | 0.0% | 0.0% | 0.0% | 0.0% |
| | | | zero-shot reasoning | 0.0% | 0.0% | 0.0% | 0.0% | 0.0% | 0.0% |
| | | DeepSeek-V3 | immediate output | 0.0% | 0.0% | 0.0% | 0.0% | 0.0% | 0.0% |
| | | | zero-shot reasoning | 0.0% | 0.0% | 0.0% | 0.0% | 0.0% | 0.0% |
| | | Qwen2.5 7B | immediate output | 0.0% | 0.0% | 0.0% | 0.0% | 0.0% | 4.0% |
| | | | zero-shot reasoning | 0.0% | 0.0% | 0.0% | 0.0% | 0.0% | 0.0% |
| | | Qwen2.5 32B | immediate output | 0.0% | 0.0% | 0.0% | 0.0% | 11.0% | 94.0% |
| | | | zero-shot reasoning | 0.0% | 0.0% | 0.0% | 0.0% | 0.0% | 0.0% |
| | | Llama-3.1-70B-Instruct | immediate output | 0.0% | 0.0% | 0.0% | 0.0% | 0.0% | 0.0% |
| | | | zero-shot reasoning | 0.0% | 0.0% | 0.0% | 0.0% | 0.0% | 0.0% |
| | | Llama-3.1 8B | immediate output | 0.0% | 0.0% | 0.0% | 0.0% | 0.0% | 0.0% |
| | | | zero-shot reasoning | 0.0% | 0.0% | 0.0% | 0.0% | 0.0% | 0.0% |
| | bucket-sort | GPT-4o mini | immediate output | 0.0% | 0.0% | 0.0% | 0.0% | 0.0% | 0.0% |
| | | | zero-shot reasoning | 0.0% | 0.0% | 0.0% | 0.0% | 0.0% | 0.0% |
| | | DeepSeek-V3 | immediate output | 0.0% | 0.0% | 0.0% | 0.0% | 0.0% | 0.0% |
| | | | zero-shot reasoning | 0.0% | 0.0% | 0.0% | 0.0% | 0.0% | 0.0% |
| | | Qwen2.5 7B | immediate output | 0.0% | 0.0% | 0.0% | 0.0% | 0.0% | 0.0% |
| | | | zero-shot reasoning | 0.0% | 0.0% | 0.0% | 0.0% | 0.0% | 0.0% |
| | | Qwen2.5 32B | immediate output | 0.0% | 0.0% | 0.0% | 0.0% | 22.0% | 1.0% |
| | | | zero-shot reasoning | 0.0% | 0.0% | 0.0% | 74.0% | 94.0% | 3.0% |
| | | Llama-3.1-70B-Instruct | immediate output | 0.0% | 0.0% | 0.0% | 0.0% | 0.0% | 0.0% |
| | | | zero-shot reasoning | 0.0% | 0.0% | 0.0% | 0.0% | 0.0% | 0.0% |
| | | Llama-3.1 8B | immediate output | 0.0% | 0.0% | 0.0% | 0.0% | 0.0% | 100.0% |
| | | | zero-shot reasoning | 0.0% | 0.0% | 0.0% | 0.0% | 0.0% | 0.0% |

Table 13: Percentage of invalid generations for Class CS languages, broken down by model, prompt template, and encoding strategy (many→one, one→one, and variants 2–5).

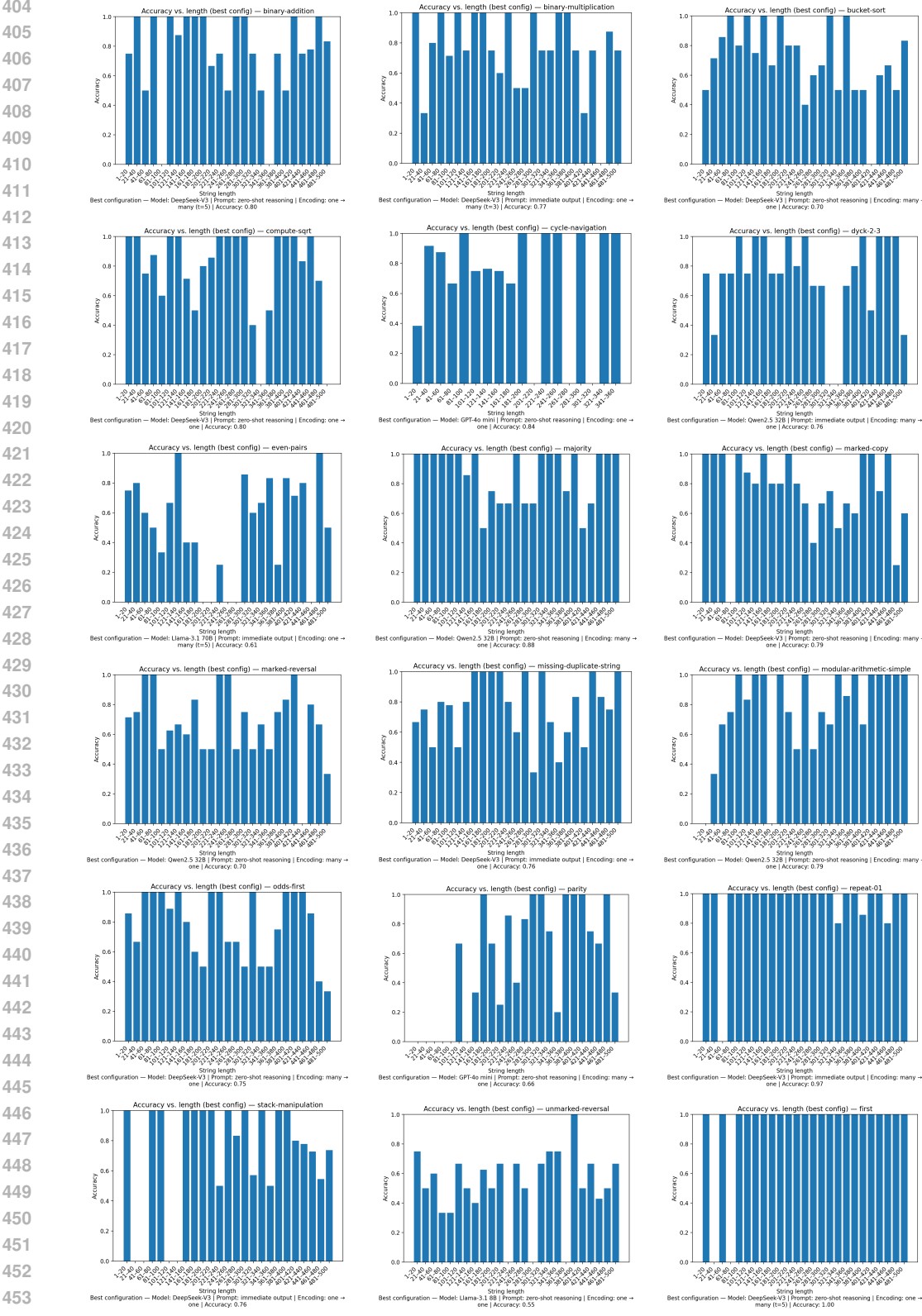

Figure 3: Accuracy histograms for all languages and best configurations.

