# OpenReview forum: "Which Formal Languages Can Large Language Models Learn in Context?"
_ICLR.cc/2026/Conference — Submitted to ICLR 2026_

### Official Review · Reviewer_opGK · 2025-10-27

**Soundness:** 2
**Presentation:** 2
**Contribution:** 2
**Rating:** 2
**Confidence:** 4

**Summary:**

This paper presents an empirical evaluation of in-context learning for formal language recognition. The authors investigate the performance of various models, including pretrained large language models and Transformers trained from scratch. The study concludes that current LLMs exhibit significant weaknesses in this domain. Furthermore, the paper explores the impact of different tokenization schemes and chain-of-thought reasoning, finding that neither strategy yields consistent accuracy improvements. The core contribution is an empirical characterization of this limitation in modern LLMs.

**Strengths:**

1. The experimental setup is applied across a diverse set of language models, including both open- and closed-source systems. This breadth strengthens the generalizability and credibility of the findings.

2. The paper identifies a task domain (formal language recognition) where a gap exists between current theoretical results and empirical performance.

**Weaknesses:**

1. Given its submission to the learning theory track, the paper's contribution would be substantially strengthened by the inclusion of theoretical analysis. The current work is purely empirical, which may not align with the expectations for this area. The authors might consider either adding theoretical analyses or reframing the paper's contribution.

2. The finding that pretrained LLMs struggle with a specialized task like formal language recognition is not unexpected. A significant limitation of the current study is the omission of experiments on finetuned models. Evaluating the in-context learning capabilities of LLMs after finetuning on this task would provide a much stronger test of their limitations and significantly enhance the paper's contribution.

3. The exposition of the string encoding strategies in Section 4.1 would benefit from significant clarification. As written, the descriptions are difficult to follow and appear to contain inconsistencies.

   - The distinction between the "one-to-one" and "many-to-one" strategies in Table 1 is unclear.

   - There appears to be an inconsistency in Section 4.1.1. Line 242 states that symbols are mapped to "distinct single-character tokens," yet the "many-to-one" example in Table 1 shows a three-character token. This also seems to conflict with the description on lines 246-247 ("potentially allowing multiple symbols to be tokenized as a single token").

4. The "one-to-many" encoding strategy (Section 4.1.3) raises methodological concerns. The introduction of random filler tokens ($\delta$) is a highly unconventional approach that diverges significantly from standard pretraining or testing procedures. It is unclear what the motivation for this strategy is, and it may introduce confounding variables that make the results difficult to interpret. The paper would be strengthened by either a clearer justification for this method or by replacing it with a more standard tokenization approach.

5. Lines 298-301 appear to have a copy-paste error: "if $y=0$ or $y=0$" and "if $y=1$ or $y=1$".

**Questions:**

1. Could the authors comment on the omission of finetuning experiments? As noted in the weaknesses, this seems to be a critical missing piece for evaluating the true capabilities of LLMs on this task.

2. Could the authors provide a thorough revision of Section 4.1 to clarify the string encoding strategies? This should include resolving the apparent inconsistencies between the text (e.g., Line 242) and the examples (Table 1).

3. What is the hypothesis behind the "one-to-many" encoding strategy? How do the authors account for the potential impact of the random filler tokens and the train-test tokenization mismatch?

---

> ### Author Response · Authors · 2025-11-20
>
> Thank you very much for your review.
>
> Weaknesses:
> > Given its submission to the learning theory track, the paper's contribution would be substantially strengthened by the inclusion of theoretical analysis. The current work is purely empirical, which may not align with the expectations for this area. The authors might consider either adding theoretical analyses or reframing the paper's contribution.
>
> While the paper does not introduce new theoretical results, our goal is precisely to shed light on existing theory by testing how recent theoretical insights about the expressive power of transformers manifest in practice. The work is therefore empirical in method but theoretical in motivation, as we use formal-language tasks to probe the kinds of structure that theoretical analyses predict transformers can or cannot capture under different conditions (e.g., padding, CoT, fixed-context computation). We agree that this makes the paper somewhat unusual for this track, but our reasoning was that rigorous empirical evaluation is necessary to assess how far current theoretical claims extend to real models and realistic prompting regimes.
>
> We agree that this connection should be made more explicit. In the revision, we will clarify the theoretical motivation behind each experimental setting and more directly relate our findings to the predictions of prior theory. We hope that this strengthens the positioning of the paper as an empirical companion to ongoing theoretical work rather than as a stand-alone empirical study.
>
> > The finding that pretrained LLMs struggle with a specialized task like formal language recognition is not unexpected. A significant limitation of the current study is the omission of experiments on finetuned models. Evaluating the in-context learning capabilities of LLMs after finetuning on this task would provide a much stronger test of their limitations and significantly enhance the paper's contribution.
>
> > Could the authors comment on the omission of finetuning experiments? As noted in the weaknesses, this seems to be a critical missing piece for evaluating the true capabilities of LLMs on this task.
>
> Thank you for this suggestion. We agree that finetuning would be a useful point of comparison. The main focus of our paper is specifically on ICL as a learning algorithm in an abstract setting. Finetuning can be seen as an alternative, albeit more expensive, learning algorithm based on LLMs, and the computational and financial cost is the main reason we did not include these experiments. We plan to run finetuning experiments on LLMs if time permits.
>
> > The exposition of the string encoding strategies in Section 4.1 would benefit from significant clarification. As written, the descriptions are difficult to follow and appear to contain inconsistencies.
>
> > > The distinction between the "one-to-one" and "many-to-one" strategies in Table 1 is unclear.
>
> > > There appears to be an inconsistency in Section 4.1.1. Line 242 states that symbols are mapped to "distinct single-character tokens," yet the "many-to-one" example in Table 1 shows a three-character token. This also seems to conflict with the description on lines 246-247 ("potentially allowing multiple symbols to be tokenized as a single token").
>
> > Could the authors provide a thorough revision of Section 4.1 to clarify the string encoding strategies? This should include resolving the apparent inconsistencies between the text (e.g., Line 242) and the examples (Table 1).
>
> Thank you for pointing this out. We will do our best to clarify these distinctions and revise Section 4.1 to remove ambiguity. Below we summarize the intended behavior.
>
> [continue in the next message]

---

> ### Author Response · Authors · 2025-11-20
>
> **One-to-one mapping.**
> Each symbol is mapped to a string that is forced to become a single token. We achieve this by prefixing each mapped symbol with a space (e.g., `" a"`, `" b"`, `" c"`). When concatenated in the prompt, these boundaries align with tokenizer boundaries, ensuring one symbol corresponds to one token.
>
> For example, the string
> $w = abbac$
>
> becomes
> ` a b b a c`
> which is tokenized into five tokens, exactly one per original symbol.
>
> **Many-to-one mapping.**
> Each symbol is mapped to a single character (e.g., `"a"`, `"b"`, `"c"`), and the resulting characters are concatenated without spaces. In this case, we do not control tokenization, and multiple characters may merge into a single token.
>
> To illustrate this, consider the same string
> $w = abbac$
>
> under many-to-one it becomes
> `abbac`.
> For GPT-4o-mini, this string is tokenized into two tokens (`"abb"` and `"ac"`), meaning the first three symbols collapse into one token and the last two into another. This behavior is exactly what the many-to-one strategy is designed to capture: tokenizer-induced merging of adjacent symbols.
>
> Thank you again for your comment; the example in Table 1 showing a multi-character token illustrates the concept, but the text did not make this mechanism explicit. We will revise the section to clearly illustrate how tokenization differs across strategies and include explicit tokenization examples to prevent confusion.
>
> > The "one-to-many" encoding strategy (Section 4.1.3) raises methodological concerns. The introduction of random filler tokens ($\delta$) is a highly unconventional approach that diverges significantly from standard pretraining or testing procedures. It is unclear what the motivation for this strategy is, and it may introduce confounding variables that make the results difficult to interpret. The paper would be strengthened by either a clearer justification for this method or by replacing it with a more standard tokenization approach.
>
> > What is the hypothesis behind the "one-to-many" encoding strategy? How do the authors account for the potential impact of the random filler tokens and the train-test tokenization mismatch?
>
> Our motivation for the one`->`many strategy is to investigate whether increasing the number of input tokens affects the model’s ability to recognize formal languages. Recent theoretical work [1] shows that transformers equipped with padding tokens can compute strictly more functions than transformers operating without padding, suggesting that the number of available "input computation steps" (i.e., tokens) can influence expressive power. We will make sure to explain this connection more clearly and reference this work appropriately in the related work section.
>
> That said, we agree with you that introducing filler tokens in this manner is unconventional but, more importantly, differs from typical pretraining settings. Our goal was to create a controlled way of increasing the effective token budget, but we acknowledge that this may introduce confounding effects.
>
> To address this, we are working on adding a more natural one`->`many encoding strategy. In particular, we will explore an approach in which each symbol is mapped to a character sequence (= a word) that is guaranteed to be tokenized into exactly $k$ tokens. For example, for a one`->`many mapping with $k = 2$, we will map each symbol to a word or string that the tokenizer consistently splits into two tokens. This will allow us to test whether any observed effects are driven by the increased number of tokens or by token-boundary artifacts introduced by our current substitution scheme.
>
> > Lines 298-301 appear to have a copy-paste error:  "if $y = 0$ or $y = 0$" and "if $y = 1$ or $y = 1$"."
>
> Thank you for your comment. This is actually not a typo, but a clarity issue in the way it is presented; it is meant to be read as $y = 0$ or $y = 0.$, where the "." is part of the string. In other words, the label can be optionally followed by a period.
>
> We will revise the text to make this explicit. For example, we can rewrite it as:
>
> \\[
> h(y) =\\; \\begin{cases}
> \\top & \\text{if } y \\in \\{1, 1.\\} \\\\
> \\bot & \\text{if } y \\in \\{0, 0.\\} \\\\
> \\emptyset & \\text{otherwise.}
> \\end{cases}
> \\]
>
> This should eliminate the ambiguity and correctly reflect the intended behavior.
>
> [1]: Merrill, William, and Ashish Sabharwal. "Exact Expressive Power of Transformers with Padding." arXiv:2505.18948 (2025).

---

> > ### Comment · Reviewer_opGK · 2025-11-25
> >
> > I thank the authors for their detailed response and for addressing several of my initial concerns.
> >
> > While I appreciate the authors' clarification that the work is "empirical in method but theoretical in motivation," a submission to the learning theory track typically requires rigorous theoretical analysis or guarantees. The absence of such analysis remains a significant weakness in this context. The motivation alone is insufficient to waive the expectation for theoretical substantiation, which is standard for this track.
> >
> > Even when evaluated strictly as an empirical contribution, the paper's scope is somewhat limited. The study focuses exclusively on pretrained LLMs; however, for the specialized task of formal language recognition, it is critical to evaluate the performance of fine-tuned models as well. Without results on finetuned LLMs, the assessment of the models' capabilities remains incomplete, limiting the broader impact of the findings.
> >
> > As the rebuttal did not include additional theoretical derivation or expand the experimental scope to include finetuning, the key limitations highlighted in my initial review remain unresolved. Therefore, I decide to maintain my original score.

---

### Official Review · Reviewer_aYf9 · 2025-10-30

**Soundness:** 3
**Presentation:** 2
**Contribution:** 2
**Rating:** 4
**Confidence:** 2

**Summary:**

This work evaluates four families of LLMs on formal language recognition task using the FLaRe benchmark, under the in-context-learning (ICL) setting. They investigated the impact of input padding length and the role of chain-of-thought. The results showed that with ICL, current LLMs are more sample-efficient than transformers trained from scratch. Increasing the input or output budget does not guarantee consistent ICL accuracy gain.

**Strengths:**

1. Originality: investigate this discrepancy between the theoretical headroom and actual performance of ICL.
2. Various experimental design to investigate the impact of input padding length and prompt strategies.
3. Extensive experiments across 4 families of LLMs of different sizes.

**Weaknesses:**

1.	In Sec 4.1, the motivation behind is that adding input padding tokens may improve performance, as stated in the abstract. It seems this point is not supported by any references.
2.	In Table 2, each accuracy is in fact the highest accuracy among all the settings (input decoding strategy, prompting strategy). The performance comparison between different models is not under the **same** setting.
3.	The result analysis seems incomplete: it lacks more fine-grained level analysis within the Chomsky hierarchy. There is no qualitative analysis.

**Questions:**

1.	I find that even for humans (such as myself), it is difficult to find the answers in Examples E.1 and E.2, because I am not aware of the rules or patterns of this specific formal language. The same applies to an LLM. Therefore, my question is: in your experiments, the LLMs were not provided with an explanation of the given formal language—under this setting, what is the implication of the model’s performance?

2.	In Sec 4.1, the one to many encoding strategy does add input computing budget, but it also makes the question more complex (e.g. the instance in Table 1, to correctly understand, the model must consider the two tokens “a” and “_p” together as a single symbol). Does this strategy introduce additional bias that may degrade model performance (i.e. more computing power but the model gets more confused)?

3.	A trivial question: the motivation about this paper includes evaluating the reasoning ability of LLMs. Why not evaluate recent large reasoning models, such as DeepSeek-R1, OpenAI o1?

4.	In Sec 5, the ICL example size is 100. Is it a design choice? Have you compared with other choices?

5.	Is there any specific reason that the authors conduct the experiments on a single FLR benchmark?

---

> ### Author Response · Authors · 2025-11-20
>
> Thank you very much for your review.
>
> > In Sec 4.1, the motivation behind is that adding input padding tokens may improve performance, as stated in the abstract. It seems this point is not supported by any references.
>
> The reference is Merrill and Sabharwal (2025) [1]. We will be sure to add this.
>
> > In Table 2, each accuracy is in fact the highest accuracy among all the settings (input decoding strategy, prompting strategy). The performance comparison between different models is not under the same setting.
>
> The purpose of Table 2 is to give a concise summary of our results and to provide an upper bound on each model’s performance by reporting the maximum score it attains across all prompting and encoding settings. Although the optimal setting may differ across models, the comparison remains fair: all models are evaluated on the same set of settings and thus have equal opportunity to achieve high accuracy under any of them. That said, we also analyze the effect of individual strategies in detail, and Appendix~G.1 reports the full per-setting breakdowns.
>
> Across settings, we observe several consistent trends. First, input compression helps: the many to one encoding yields the highest accuracies for most models, whereas one to many encodings often degrade performance as the expansion factor increases. Second, the effect of prompting strategy is mixed: allowing chain-of-thought reasoning produces only modest gains at best and often reduces accuracy relative to the immediate-output prompt. Third, model scale matters: larger or proprietary LLMs models reliably outperform smaller ones.
>
> We would be happy to include additional analyses or alternative aggregations if you have particular comparisons in mind. For example, we could report a variance-based analysis, quantifying how sensitive each model’s performance is to the choice of prompting or encoding strategy. Concretely, for each model--language pair, we can compute the variance (or range) of accuracies across all settings; high variance would indicate sensitivity to the choice of strategy, while low variance would suggest robustness.
>
> > The result analysis seems incomplete: it lacks more fine-grained level analysis within the Chomsky hierarchy. There is no qualitative analysis.
>
> Thank you for pointing this out. We agree that additional analysis could strengthen the presentation. We can elaborate on contrasts among classes, comparisons within a specific class, or more detailed case studies for individual languages.
>
> Questions:
> > I find that even for humans (such as myself), it is difficult to find the answers in Examples E.1 and E.2, because I am not aware of the rules or patterns of this specific formal language. The same applies to an LLM. Therefore, my question is: in your experiments, the LLMs were not provided with an explanation of the given formal language—under this setting, what is the implication of the model’s performance?
>
> The purpose of our work is to test the abstract reasoning abilities of LLMs by testing how well they infer rules from in-context examples, without being given explicit rules. This is similar to any supervised learning setup, where in our case the learner is an LLM instead of a network trained from scratch. Although any finite training set is consistent with multiple possible languages, each of the languages in the FLaRe benchmark has a fairly simple underlying rule that can be inferred using Occam's razor, so LLMs and humans should converge to the same solution. Granted, the example you cited, Binary Addition, is one of the more difficult languages in the benchmark, being in the context-sensitive class. LLMs' generally poor performance on these tasks implies that their ICL mechanism does not infer the right rules, or that the LLM cannot reason with those rules, or some combination of both. So, in practice they may be relying on more surface-level heuristics to perform ICL rather than deep algorithmic reasoning.
>
> [continue in the next message]

---

> ### Author Response · Authors · 2025-11-20
>
> > In Sec 4.1, the one to many encoding strategy does add input computing budget, but it also makes the question more complex (e.g. the instance in Table 1, to correctly understand, the model must consider the two tokens `a` and `_p` together as a single symbol). Does this strategy introduce additional bias that may degrade model performance (i.e. more computing power but the model gets more confused)?
>
> Thank you for raising this point. We agree that the current one to many encoding strategy could introduce unintended biases: expanding a single symbol into an arbitrary sequence of tokens may make the task more complex and force the model to track token groupings that do not correspond to natural tokenization patterns.
>
> To address this possibility, we plan to include additional experiments using an alternative one to many strategy in which each symbol is mapped to a character sequence (= a word) that is guaranteed to be tokenized into exactly $n$ tokens. For example, for $n = 2$, we will map each symbol to a carefully chosen string that the tokenizer consistently splits into two tokens. This will allow us to test whether the degradation we observe is due to the increased number of tokens alone or to the specific token-boundary artifacts introduced by the current scheme.
>
> We will report these results in the revised version and discuss their implications.
>
> > A trivial question: the motivation about this paper includes evaluating the reasoning ability of LLMs. Why not evaluate recent large reasoning models, such as DeepSeek-R1, OpenAI o1?
>
> We agree that including recent large reasoning models would be interesting and could further broaden the empirical picture. For instance, it would allow us to compare how pretrained LLMs using CoT prompting differ from LRMs whose reasoning abilities are explicitly trained into the model. The reason we have excluded reasoning models is that we wanted to compare the performance with and without CoT for the *same* model, which we have done by tweaking the instructions in the prompt to non-reasoning models. Reasoning models are post-trained always to use CoT in the backend, so as the end user we don't have a way to turn CoT off for a direct comparison.
>
> We see evaluating models such as DeepSeek-R1 or OpenAI o1 as a natural extension of this work, and we plan to explore this direction in future experiments. If there are specific models or evaluation settings you believe would align well with our framework, we would be happy to consider them.
>
> > In Sec 5, the ICL example size is 100. Is it a design choice? Have you compared with other choices?
>
> We chose 100 examples because it is roughly the largest number of examples that fits within the context window of all of the LLMs we evaluate. We also experimented with smaller in-context example sizes on a subset of models (namely the open-source ones), and we would be happy to include these results in the revised version.
>
> > Is there any specific reason that the authors conduct the experiments on a single FLR benchmark?
>
> FLaRe is the only formal language recognition benchmark we're aware of that spans multiple classes of the Chomsky hierarchy. A similar benchmark is MLRegTest, but it only includes regular languages. We chose to focus on a single benchmark that has a higher diversity of language types. If you have suggestions for additional benchmarks that you believe would fit these criteria, we would be happy to consider them.
>
> [1]: Merrill, William, and Ashish Sabharwal. "Exact Expressive Power of Transformers with Padding." arXiv preprint arXiv:2505.18948 (2025).
>
> [2]: Akyürek, Ekin, et al. MLRegTest: A benchmark for regular-language generalization in large neural networks. arXiv preprint arXiv:2304.07687 (2023).

---

> > ### Comment · Reviewer_aYf9 · 2025-11-27
> >
> > I thank the authors for the clarification. Hope some of the points can be incorporated into the revised version to further improve the paper.

---

### Official Review · Reviewer_o9G9 · 2025-10-30

**Soundness:** 4
**Presentation:** 3
**Contribution:** 3
**Rating:** 6
**Confidence:** 5

**Summary:**

The authors evaluate the performance of ICL on a variety of existing formal language benchmarks that capture canonical problems from various levels of the Chomsky hierarchy. They obfuscate these problems via symbol permutation and vary both symbol encoding and whether the models are acting in chain-of-thought. They find that these LLMs cannot perfectly solve all but one of the tasks, and in general, substantially underperform the best performance an even much smaller transformer can provide if trained on the underlying data distribution. They find that, however, this ICL generally outperforms training a transformer from scratch on just the examples provided to ICL. Additionally, they find, in contradiction of high level heuristics commonly believed in the field of LLM prompting, that on these tasks COT has limited effect, and that reducing the number of tokens in the input appears to improve performance generally.

**Strengths:**

The overall approach of this paper is precise and directed at solving the problem it poses. It has a clear, formal description of both the problem and experiments.

The comparison to both a small transformer train from scratch on the exact data and a larger sample representing the data distribution is interesting and provides two distinctly helpful points of comparison.

One -> many and many -> one are interesting points of comparison.

**Weaknesses:**

In the abstract and introduction, it is stated that “ICL with pretrained LLMs is consistently more accurate than training a small transformer from scratch on the same data” and “pretrained LLMs are markedly more sample-efficient than transformers trained from scratch on the same data.” The second of these accurately and precisely describes the result, however, the first of these has the (probably unintentional) effect of implying that the small transformers are trained in a standard way on the same data distribution, rather than trained in an unusual way (with only 100 samples) on the same exact data points. The fact that training transformers from scratch with many examples outperforms ICL should also be discussed more prominently in the introduction, to properly contextualize this finding. Training on 100 examples is extremely unusual and it is unsurprising this performs quite poorly. [To be clear, I acknowledge that situations with only 100 examples are common, motivating the use of ICL and making this a valid comparison to include in the paper; this weakness is entirely in phrasing and the omission of the 10000 sample experiment from the introduction.]

No reasoning models are included in this evaluation, and the only proprietary model included is the relatively weak gpt-4o-mini. The inclusion of gpt-5-mini or o3-mini might be helpful as a point of reference.

A baseline that might be useful is something similar to Ayurek 2024 [1]. This involves training models on the structure of formal language ICL tasks. You might do that, or perhaps just train a small transformer on formal language tasks in general rather than one specific task, then fine-tune on one specific task. This would allow you to isolate the degree to which 100 examples are insufficient for a small transformer to learn the language, or whether training a transformer from scratch requires more than 100 examples in general.

Minor:
113-116: Gupta 2025 does investigate recognition via the Transducer task (where the task is to classify a sequence as belonging to a language or not based on its prefixes); the more relevant difference seems to be the difference in languages (randomly sampled regular languages vs a richer but non-randomly sampled set of more canonical languages from throughout the Chomsky Hierarchy). This means “previously unstudied angle: in-context formal language recognition” from earlier in the introduction should be rephrased.

A discussion of [1] should be included in related work.

Tables: a lack of error bars on the numbers makes it difficult to determine whether certain comparisons are meaningful or not.

[1]: Akyürek, Ekin, et al. "In-context language learning: Architectures and algorithms." arXiv preprint arXiv:2401.12973 (2024).

**Questions:**

The result about many -> one being superior to one -> one and one -> many is striking; do you have any hypotheses on why this might be the case? I have seen some prior work with similar (though unsystematic) results on how “better” tokenization makes this kind of task worse (e.g., the Commas appendix of Gupta 2025), but have never found a compelling reason as to why.

---

> ### Author Response · Authors · 2025-11-20
>
> Thank you very much for your detailed and constructive review!
>
> > In the abstract and introduction, it is stated that "ICL with pretrained LLMs is consistently more accurate than training a small transformer from scratch on the same data"...
>
> Thank you for pointing this out; we will rephrase this to make it clearer that this result only applies when the transformer is trained on the same amount of training data.
>
> > No reasoning models are included in this evaluation, and the only proprietary model included is the relatively weak gpt-4o-mini. The inclusion of gpt-5-mini or o3-mini might be helpful as a point of reference.
>
> Thank you for these suggestions; we will add additional models if we have time. The reason we have excluded reasoning models is that we wanted to compare the performance with and without CoT for the *same* model, which we have done by tweaking the instructions in the prompt to non-reasoning models. Reasoning models are post-trained always to use CoT in the backend, so as the end user we don't have a way to turn CoT off for a direct comparison.
>
> Are there particular non-reasoning proprietary models stronger than gpt-4o-mini that you would suggest using, so that we can do a controlled comparison with and without CoT?
>
> > A baseline that might be useful is something similar to Ayurek 2024 [1]...
>
> Thank you for the suggestion; we agree that a meta-learning ICL setup in the style of Akyürek et al. (2024) would be a useful point of comparison.
>
> Setting this up would require hashing out several important details, so we may not have time to execute these experiments during the rebuttal, although we are still open to discussing it further and providing some preliminary results. For one, this meta-learning setup requires defining a *distribution* over a family of languages. Our paper uses a fixed set of hand-picked languages, so we would need to work out which distribution of languages is appropriate for the meta-training step. The results will be sensitive to the particular choice of distribution over meta-tasks, so this decision is important; for example, it would not make sense to meta-train on regular languages and expect a model to succeed on non-regular languages. The main scientific question we are exploring in our paper is whether pre-training on massive corpora of natural language effectively meta-trains the LLM to do ICL of formal languages, which would give evidence of abstract reasoning abilities. Is there a particular family/distribution of artificial meta-tasks that you had in mind?
>
> Note that Grau-Moya et al. (2024) [2] ran a meta-learning study that is relevant to this discussion using randomly-sampled programs from a Turing-complete programming language.
>
> > Minor: 113-116: Gupta 2025 does investigate recognition via the Transducer task...This means "previously unstudied angle: in-context formal language recognition" from earlier in the introduction should be rephrased.
>
> Their setup is a bit different from ours, but you are right that their way of encoding the input is essentially a more compact way of providing ICL exemplars to the model, where the exemplars happen to be prefixes of the same string. We will update the text to acknowledge this.
>
> > A discussion of [1] should be included in related work.
>
> We will add this.
>
> > Tables: a lack of error bars on the numbers makes it difficult to determine whether certain comparisons are meaningful or not.
>
> Due to the cost of running our experiments, we ran just one trial. If we have time, we will work on running additional experiments and reporting the variance across multiple runs.
>
> > The result about many `->` one being superior to one `->` one and one `->` many is striking; do you have any hypotheses on why this might be the case?...
>
> One hypothesis is that many-to-one corresponds more closely to the way that the model is pre-trained on natural language, i.e., with multiple characters packed into the same token. When processing natural language, the model often needs to rely on sub-token information, so it probably has an internal mechanism for unpacking tokens into their constituent characters. Even though additional tokens result in greater theoretical expressivity, the most important factor in practice is probably whether the task we are evaluating matches the pre-training task.
>
> [1] Akyürek, Ekin et al. "In-context language learning: Architectures and algorithms." 2024. URL: https://proceedings.mlr.press/v235/akyurek24a.html
>
> [2] Grau-Moya, Jordi et al. "Learning Universal Predictors." 2024. URL: https://proceedings.mlr.press/v235/grau-moya24a.html

---

> > ### Comment · Reviewer_o9G9 · 2025-11-22
> >
> > Thank you for responding to many of the points we raised! There are a few outstanding concerns that remain. I also apologize for completely missing that you did in fact cite Akyurek 2024 in your related work.
> >
> > > Thank you for these suggestions; we will add additional models if we have time. The reason we have excluded reasoning models is that we wanted to compare the performance with and without CoT for the same model, which we have done by tweaking the instructions in the prompt to non-reasoning models. Reasoning models are post-trained always to use CoT in the backend, so as the end user we don't have a way to turn CoT off for a direct comparison.
> > > Are there particular non-reasoning proprietary models stronger than gpt-4o-mini that you would suggest using, so that we can do a controlled comparison with and without CoT?
> >
> > Your point regarding with/without CoT is fair and I missed this as a potential reason for this model selection. I think an acknowledgement that this is the reason that reasoning or larger proprietary models are not included belongs in the paper, even as a footnote.
> >
> > > Thank you for the suggestion; we agree that a meta-learning ICL setup in the style of Akyürek et al. (2024) would be a useful point of comparison….
> >
> > This is a fair point regarding the inherent complexity of this suggestion.
> >
> > > Due to the cost of running our experiments, we ran just one trial. If we have time, we will work on running additional experiments and reporting the variance across multiple runs.
> >
> > Perhaps aggregating across the tasks you already have might work as something in the interim? In my view, some level of evidence that comparisons are statistically meaningful is a requirement.
> >
> > > One hypothesis is that many-to-one corresponds more closely to the way that the model is pre-trained on natural language, i.e., with multiple characters packed into the same token. When processing natural language, the model often needs to rely on sub-token information, so it probably has an internal mechanism for unpacking tokens into their constituent characters. Even though additional tokens result in greater theoretical expressivity, the most important factor in practice is probably whether the task we are evaluating matches the pre-training task.
> >
> > This is an interesting hypothesis, thank you for sharing!

---

### Official Review · Reviewer_Dj3D · 2025-10-31

**Soundness:** 3
**Presentation:** 2
**Contribution:** 2
**Rating:** 2
**Confidence:** 3

**Summary:**

This paper studies the ICL capabilities of LLMs in learning formal languages. By considering different string encoding and prompting strategies, the authors conduct extensive empirical experiments across the Chomsky hierarchy. The results suggest that the success of ICL in recognizing formal languages stems from better approximation rather than greater computational power.

**Strengths:**

- The experiments are extensive, covering various string encoding strategies, prompting methods, languages across the Chomsky hierarchy, and multiple models.

- The findings provide useful insights into the mechanism of LLM ICL, suggesting that LLMs might not perform actual computation and learning.

**Weaknesses:**

- Although the empirical study is extensive, the paper lacks in-depth analysis of the results. For instance, while different models show varying performance across languages, the paper does not discuss why a certain model outperforms others for one language but underperforms for another.

- The paper does not offer a theoretical explanation for the mechanisms underlying the observed empirical results.

**Questions:**

See Weaknesses.

---

> ### Author Response · Authors · 2025-11-20
>
> Thank you very much for your review! We look forward to engaging with you to address the issues you raised.
>
> >Although the empirical study is extensive, the paper lacks in-depth analysis of the results. For instance, while different models show varying performance across languages, the paper does not discuss why a certain model outperforms others for one language but underperforms for another.
>
> We agree that some of the cross-model and cross-language performance differences merit further discussion. We are currently investigating several model-specific factors---such as differences in post-training procedures, positional encodings, architectural choices (e.g., pre-norm vs. post-norm), and the context lengths used during training---that may influence how reliably a model captures the structural properties of the target languages, namely the rules that characterize their well-formed strings. We will incorporate a more detailed analysis of these aspects in the revised version. Are there any specific hypotheses or factors you believe would be valuable to explore?
>
> >The paper does not offer a theoretical explanation for the mechanisms underlying the observed empirical results.
>
> Our work is chiefly empirical rather than theoretical, but it is directly motivated by existing theoretical results [1, 2, 3] that characterize what kinds of computations transformer models can or cannot perform under different conditions, including chain-of-thought (CoT) reasoning in the output and the use of padding tokens in the input. Our goal is to investigate how these theoretical insights manifest in practice. Specifically, we aim to study the expressivity of ICL, not only by testing whether LLMs can perform particular formal-language computations, but also by controlling the presence of padding tokens in the input and CoT tokens in the output.
>
> We agree that a deeper theoretical interpretation of the mechanisms underlying our empirical findings would be valuable. In the revision, we will expand the discussion section to make the connection between theory and empirical observations more explicit. If there are particular theoretical angles, analyses, or additional experiments that you believes would help clarify these mechanisms, we would greatly appreciate your suggestions.
>
> [1] Jiaoda Li and Ryan Cotterell. Characterizing the expressivity of transformer language models, 2025.
>
> [2] William Merrill and Ashish Sabharwal. Exact expressive power of transformers with padding, 2025.
>
> [3] William Merrill and Ashish Sabharwal. The expressive power of transformers with chain of thought. In ICLR 2024.

---

> > ### Comment · Reviewer_Dj3D · 2025-11-28
> >
> > I thank the authors for their detailed responses, which address part of my concerns.
> >
> > This work presents interesting empirical results. However, I agree with Reviewer opGK that a purely empirical study with theoretical motivation is insufficient for the learning theory track. As a result, I am changing my score to 4 and leaving it to the area chair to judge whether the paper is suitable for publication under this track.

---

### Meta-Review · Area_Chair_tHJj · 2026-01-05

**Summary:**

The authors have studied in-context learning (ICL) over several formal language benchmarks representing problems across different levels of the Chomsky hierarchy. They control the input token budget by changing how many tokens encode each symbol and the output token budget by comparing prompts that require immediate output to prompts that allow intermediate reasoning. They have shown that increasing input or output token budgets does not improve ICL accuracy and pretrained LLMs are more sample-efficient than transformers trained from scratch on the same data distribution.


The paper was thoroughly discussed by our reviewers. The reviewers have major concerns such as lack of theoretical results (Reviewer Dj3D and Reviewer opGK) and more fine-grained level analysis within the Chomsky hierarchy (Reviewer aYf9). I recommend the authors to carefully address reviewers’ comments and improve the paper for future resubmissions

**Reviewer Concerns:**

Some comments have been well addressed such as "No reasoning models are included in this evaluation, and the only proprietary model included is the relatively weak gpt-4o-mini. The inclusion of gpt-5-mini or o3-mini might be helpful as a point of reference." and "In Table 2, each accuracy is in fact the highest accuracy among all the settings (input decoding strategy, prompting strategy). The performance comparison between different models is not under the same setting".

There are still major outstanding comments such as lack of theoretical results (Reviewer Dj3D and Reviewer opGK) and more fine-grained level analysis within the Chomsky hierarchy (Reviewer aYf9).

**Reviewer Scores:**

It seems to me that the majority of reviewers (Reviewer Dj3D, Reviewer o9G9, and Reviewer opGK) would have kept their scores unchanged if they had been able to participate fully in the discussion with the exception of Reviewer aYf9.

---

### Decision · Program_Chairs · 2026-01-26

Reject